# Biomechanics and neural circuits for vestibular-induced fine postural control in larval zebrafish

Takumi Sugioka [1,2,3], Masashi Tanimoto [1,2,3,4] & Shin-ichi Higashijima [1,2,3,4]

Land-walking vertebrates maintain a desirable posture by finely controlling muscles. It is unclear whether fish also finely control posture in the water. Here, we showed that larval zebrafish have fine posture control. When roll-tilted, fish recovered their upright posture using a reflex behavior, which was a slight body bend near the swim bladder. The vestibular-induced body bend produces a misalignment between gravity and buoyancy, generating a moment of force that recovers the upright posture. We identified the neural circuits for the reflex, including the vestibular nucleus (tangential nucleus) through reticulospinal neurons (neurons in the nucleus of the medial longitudinal fasciculus) to the spinal cord, and finally to the posterior hypaxial muscles, a special class of muscles near the swim bladder. These results suggest that fish maintain a dorsal-up posture by frequently performing the body bend reflex and demonstrate that the reticulospinal pathway plays a critical role in fine postural control.

Maintaining posture is important for survival in many animals. Deviation from desirable body orientation evokes corrective movements to recover the original orientation. When the posture is disturbed, two types of postural correction movements occur in land-walking vertebrates. The first type is vigorous movements (dynamic control); when deviations are large, animals correct their posture by performing stepping or walking[1]. The second type is fine movements (static control); upon small deviations, animals correct their posture by adjusting the contractions of anti-gravity muscles[2]. Because the body slightly sways all the time, this fine control is continuously working in land-walking vertebrates when they are standing[3].

Vestibular information plays an important role in the neural mechanisms that control posture[4,5]. The head orientation relative to gravity is received by the otolith organ in the inner ear. The vestibular signal (i.e., tilt) is transmitted to the vestibular nuclei in the hindbrain, and ultimately, the motor commands are sent to the spinal cord. Numerous studies have been conducted to identify the neuronal pathways that are involved in postural controls. These studies highlight the importance of vestibulo-spinal neurons, which convey the vestibular signal from the vestibular nuclei directly to the spinal cord[6–8]. In addition to this direct pathway, it is presumed that indirect pathways from the vestibular nuclei through reticulospinal (RS) neurons to the spinal cord also play a role in the vestibular-induced postural responses[9–12]. In most cases, however, conclusive evidence for the functional importance of the indirect pathways is lacking except for a limited number of recent studies[13].

The larval zebrafish is a useful animal model to study neuronal circuits for vestibular-dependent postural control. The vestibular system is largely conserved among vertebrates[14–16]. By 5 days post-fertilization, larvae start maintaining a dorsal-up posture[17], indicating that neuronal circuits for postural control are functional by this stage. The transparent brain enables functional imaging in the animals that are subjected to vestibular stimuli[18–20].

For postural correction movements, larval zebrafish can recover from a roll-tilted posture by performing swimming[21], similar to other fish such as lamprey[22]. In contrast to this dynamic postural control, it

[1]Exploratory Research Center on Life and Living Systems, Okazaki, Aichi 444-8787, Japan. [2]National Institute for Basic Biology, Okazaki, Aichi 444-8787, Japan. [3]The Graduate University for Advanced Studies, SOKENDAI, Okazaki, Aichi 444-8787, Japan. [4]These authors jointly supervised this work: Masashi Tanimoto, Shin-ichi Higashijima. ✉e-mail: tanimoto@nibb.ac.jp; shigashi@nibb.ac.jp

remains unclear whether larval zebrafish (and other fish) possess a fine postural control mechanism to recover from roll-tilted posture. A study showed that an artificial otolith displacement induces a body bend reflex that does not accompany swimming[23]. This body bend reflex may be a fine postural correction behavior. However, how the behavior contributes to postural correction remains unknown. Moreover, the neuronal circuits that control this behavior remain unclear.

Here, we examined biomechanics and neural circuits for fine postural control in larval zebrafish. We first showed that without swimming, fish correct their roll-tilted postures using a reflex behavior, which is a slight body bend near the swim bladder. Thus, this study reveals the physiological role of the bend reflex that was previously reported[23]. We also provide a physical model for the postural correction mechanism; the bend produces a misalignment between gravity and buoyancy, generating a moment of force with a rotational direction toward recovering the upright posture. We then revealed the neuronal circuits that control the reflex, including the tangential nucleus (a vestibular nucleus) through neurons in the nucleus of the medial longitudinal fasciculus (a class of RS neurons) to the spinal cord, and finally to the posterior hypaxial muscles. This study highlights the importance of indirect postural control pathways in vertebrates.

## Results

### Larval zebrafish correct their roll-tilted posture by performing the body bend reflex

To examine whether larval zebrafish possess fine postural control mechanisms, we first observed how fish responded to postural perturbation in the roll axis. A larva was placed in a small chamber, and its behavior was imaged from the frontal and dorsal sides (Fig. 1a, Supplementary Fig. 1a). To focus on mechanisms that were independent on visual information, experiments were performed in a dark room using infrared light illumination. During a leftward tilt of the chamber (Fig. 1a), the larva was also tilted in the left-downward direction due to the viscosity of the surrounding water (Fig. 1b, middle top). The fish returned to the original posture within a few seconds (Fig. 1b, right top; Supplementary Movie 1; note that the dorsal view of the fish became oblique, not vertical, after the postural correction due to rotation of

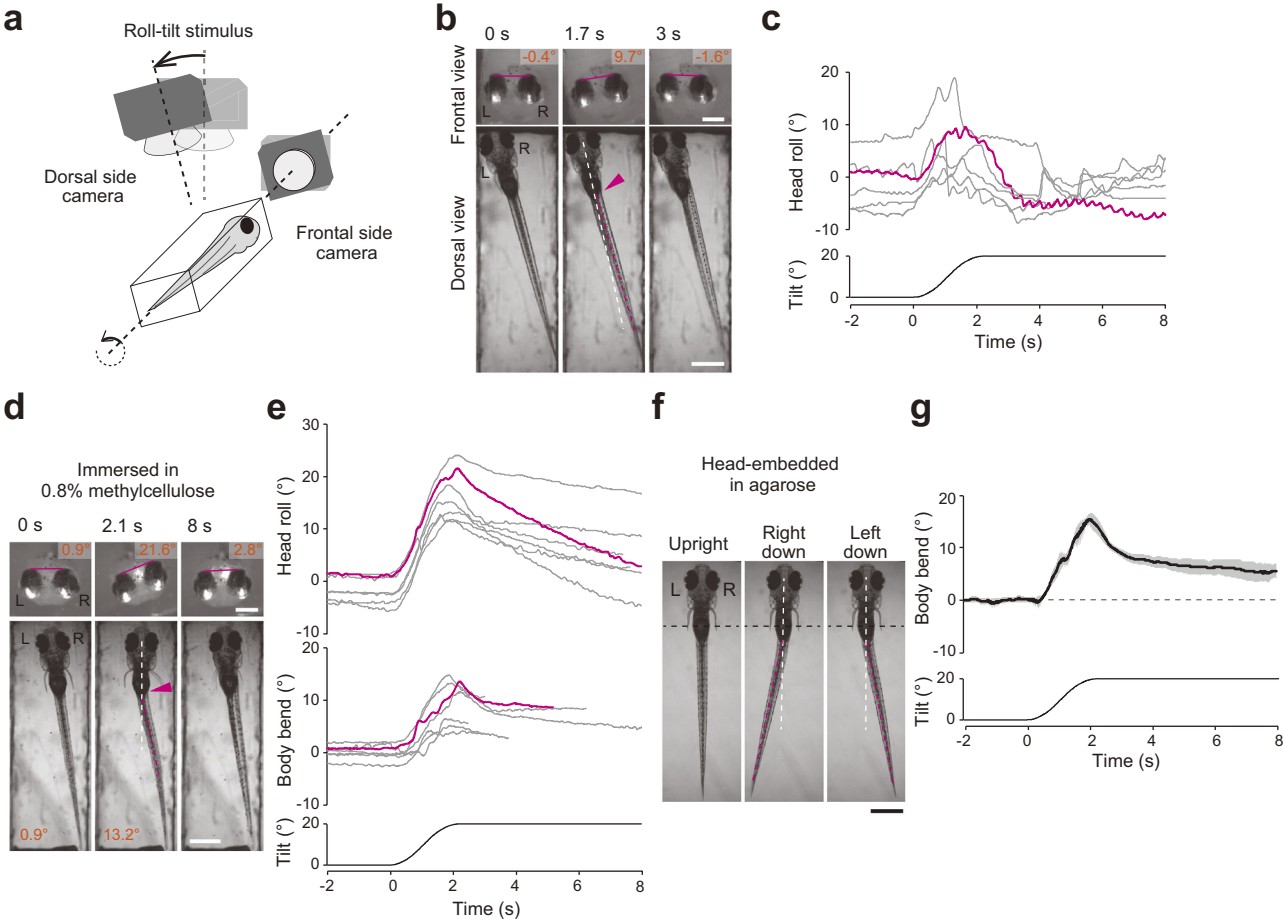

**Fig. 1 | Larval fish perform the vestibular-induced bend reflex (VBR) during roll tilt. a** Schematic illustration of the behavioral imaging. **b** Snapshots of the fish during a left-down tilt. The frontal images are horizontally flipped (mirror-imaged) such that the left–right relationship matches with that of the dorsal images. The bars in the frontal images show the lines connecting the tops of the eyes. Head-roll angles are indicated at the top. The white dashed line denotes the midline of the rostral region of the fish. The magenta dashed line denotes the line connecting the caudal end of the swim bladder and the tail end. The magenta arrowhead indicates a body bend. The dorsal view of the fish became oblique when the fish recovered the posture (3 s) due to rotation of the dorsal side camera. **c** Time course of the head roll angles in response to tilt stimuli. Six trials in a fish are shown. The magenta trace corresponds to the trial shown in **b. d, e** Same as **b** and **c**, except fish were in water

containing 0.8% methylcellulose. The body bend angles are shown in the bottom panels in **d**. In the middle panel in **e**, traces of the body bend angles are shown. Traces are terminated when the difference in the tilt angles between the chamber (bottom) and the head (top) exceeded 10° (see Methods). Seven trials from five fish are shown. The magenta traces correspond to the trial shown in **d. f** Snapshots of the head-embedded fish imaged from the dorsal side. In the middle and right panels, images of the fish with the maximum bends are shown. The black dashed lines indicate the edge of the agarose. **g** Time course of the body bend angles of a fish in response to tilt stimuli. The average and standard deviation of five trials from one fish are shown using the black line and gray shade, respectively. Scale bars, frontal images in **b, d** 200 μm; dorsal images in **b, d, f** 500 μm. Source data are provided as a Source Data file.

the dorsal side camera). Fish occasionally performed swimming to correct the perturbed posture during the trials. To focus on fine postural control mechanisms, we only collected trials in which the fish did not perform swimming. The peak of the head roll was observed at approximately 1.7 s under our experimental conditions (Fig. 1c). We measured the angular velocity of the corrective counter-roll movement when the movements were prominent (a 1.5-s time window starting at the peak of the head roll). The median value was −7.2°/s (Supplementary Fig. 1g). The counter-roll movements often overshot the upright position, and in many trials, the head roll angles were below the original values (Fig. 1b, c, Supplementary Fig. 1d; median value at 0 s was −3.0°, while that at 4 s was −7.4°). These results indicate that fish have the capability to recover from the roll-tilted posture without performing swimming.

During the postural recovery, rhythmic pectoral fin movements were often observed (Supplementary Movie 1). To examine whether the pectoral fin movements were the main cause for the postural recovery, the same experiments were conducted on fish that had their pectoral fins amputated. The fin-removed fish recovered the upright posture in most of the trials (Supplementary Fig. 1b, c, e; Supplementary Movie 2). The angular velocity amplitude of the counter-roll movement was slightly decreased compared with that of the intact fish (Supplementary Fig. 1g, median values, −7.2°/s for the intact fish compared with −6.3°/s for the finless fish, $p = 0.04$). Additionally, the overshooting counter-roll movements were not obvious (Supplementary Fig. 1e, median value, −0.4° at 0 s, and −1.5° at 4 s) and the roll angles at 4 s of fin-removed fish were larger than those of intact fish (Supplementary Fig. 1h, $p = 0.0005$). The slight decrease of the angular velocity amplitude and disappearance of the overshoot suggests that the counter-roll movements became less powerful, and thus, the pectoral fins may partially contribute to the postural recovery. However, the results strongly suggest that the fine postural control in the roll axis is mainly achieved by mechanisms without the pectoral fins.

Upon close examinations of the dorsal side images, we noticed that the body was slightly bending to the ear-up direction near the swim bladder during the recovery phase in both the intact and fin-removed fish (magenta arrowheads in Fig. 1b and Supplementary Fig. 1b; Supplementary Movies 1, 2). We speculated that this body bending contributes to the postural recovery. The bending angle of the body was small, and not highly obvious. We speculated that this is probably because the fish continuously corrected the perturbed posture using the small bend.

To observe the body movement more clearly, we performed the experiments in water containing 0.8% methylcellulose. The expectation was that the high viscosity of methylcellulose solution would slow the postural recovery process, enabling clearer observation of the body movements. The results showed that the postural recovery process was greatly slowed. Upon roll tilt of the chamber, the head roll angle showed a clear increase with its maximum sometimes reaching the maximum tilt angle of the chamber (20°) at ~2 s, which corresponded to the end timepoint of the chamber tilt (Fig. 1d, middle top; Fig. 1e). Then, the head roll angle slowly returned toward the original angle between 2 s and 8 s (Fig. 1d, top; Fig. 1e, top; Supplementary Movie 3). The body bent near the swim bladder toward the ear-up direction (arrowhead in Fig. 1d; Supplementary Movie 3). The angle of the body bend increased as the chamber tilted and then slowly decreased as the fish returned to the upright position. The bending consistently occurred in all the trials (Fig. 1e, middle). The results showed that the body bend reflex is associated with the fine postural recovery. In this paper, this behavior is referred to as the vestibular-induced bend reflex (VBR).

As noted above, the degree of the VBR showed a temporal correlation with the head roll angle (Fig. 1e, top and middle). This suggested that the VBR would be pronounced or persistent under conditions where the fish were unable to recover from a tilted posture.

We tested this idea by applying the roll-tilt stimuli to larvae with their heads embedded in agarose. The behavior of the fish during the roll tilt was observed using the apparatus shown in Supplementary Fig. 1i. As expected, the VBR was observed when the fish was tilted (Fig. 1f; similar behaviors were observed in the artificial otolith displacement experiments[23]). The tail deflected to the right during the left-down tilt, whereas it deflected to the left during the right-down tilt (Fig. 1f; Supplementary Movie 4). The body bend angle increased as the head tilt angle increased. When the tilt angle reached a constant value (20°), the bend angle began to decrease. However, the angle was kept above 0°, as expected (Fig. 1g). VBR was observed in all the fish tested, although the body bend angle varied across trials and fish examined (Supplementary Fig. 1j).

Our results thus far strongly suggest that VBR is the behavior that corrects a roll tilt. Then, how does the VBR produce the force that recovers upright posture from the roll tilt? We built a simplified model using two forces, as described below. Gravity acts at the center of mass (COM), while buoyancy acts at the center of volume (COV). In larval fish, both the COM and COV are located near the swim bladder[24]. When a fish is in the upright posture without the VBR, the fish is in an equilibrium state in the roll axis. In this situation, the COM and COV are on the midline, and gravity and buoyancy are antiparallel with the same strength on the same axis (Fig. 2a, left bottom). Upon the roll tilt, a fish performs the VBR with the bend near the swim bladder. This bend deflects head and caudal body toward ear-up side, and in reaction, the body around the swim bladder moves toward the ear-down side (Fig. 2a, right top and middle). With this dogleg-shaped body bend, the positions of the COM and COV change in the body. In the cross-section near the swim bladder around where both the COM and COV are located, the COM and COV are no longer on the midline, and both move toward the ear-up side. However, the degree of the shifts is not equal. This is because the density of the gas-filled swim bladder is smaller than that of the rest of the body. Due to this large difference in density, the COM shifts more laterally than the COV (Fig. 2a, right bottom; for more details, see Supplementary Fig. 1k, l and Discussion). This results in a misalignment between gravity and buoyancy, generating a moment of force that counter-rotates the tilted body to the upright posture (Fig. 2a, right bottom).

If the model described above is correct, swim bladder-deflated fish would not be able to recover from the tilted posture because in this condition, the density of the body has become nearly uniform, and thus, gravity and buoyancy remain aligned on the same axis regardless of the VBR (Fig. 2b, bottom). To verify this idea, we conducted behavioral experiments using fish with deflated swim bladders. To compensate for the decrease in buoyancy in the fish, experiments were performed in water that contained 12.5% sucrose. As expected, the fish was unable to recover the upright posture in most of the trials (Fig. 2c; Fig. 2d, top; Supplementary Movie 5). Population data (Supplementary Fig. 1f) showed that the median head roll angle at 4 s was 15.9°, which was not much different from that at 1.7 s (15.2°). Additionally, the head roll angles at 4 s of swim bladder-deflated fish were significantly larger than those of intact fish (Supplementary Fig. 1h; $p = 4.8 \times 10^{-6}$). These results indicate that the swim bladder-deflated fish lost the ability to recover from the roll-tilted posture. We also found that the VBR continued during the entire period when the fish was roll-tilted (arrowheads in Fig. 2c; Fig. 2d, middle; Supplementary Movie 5). This was probably because swim bladder-deflated fish were unable to recover from the tilted posture, similar to the fish that had its head embedded in agarose. Taken together, these results demonstrate that fish have a fine control mechanism to recover the upright posture in the roll axis without swimming, which is that fish correct their roll-tilted posture by performing the VBR.

## TAN−nMLF−PHM pathway may produce the VBR

Next, we explored the neural circuits that are responsible for the VBR during the roll tilt. Previous studies showed that neurons in the

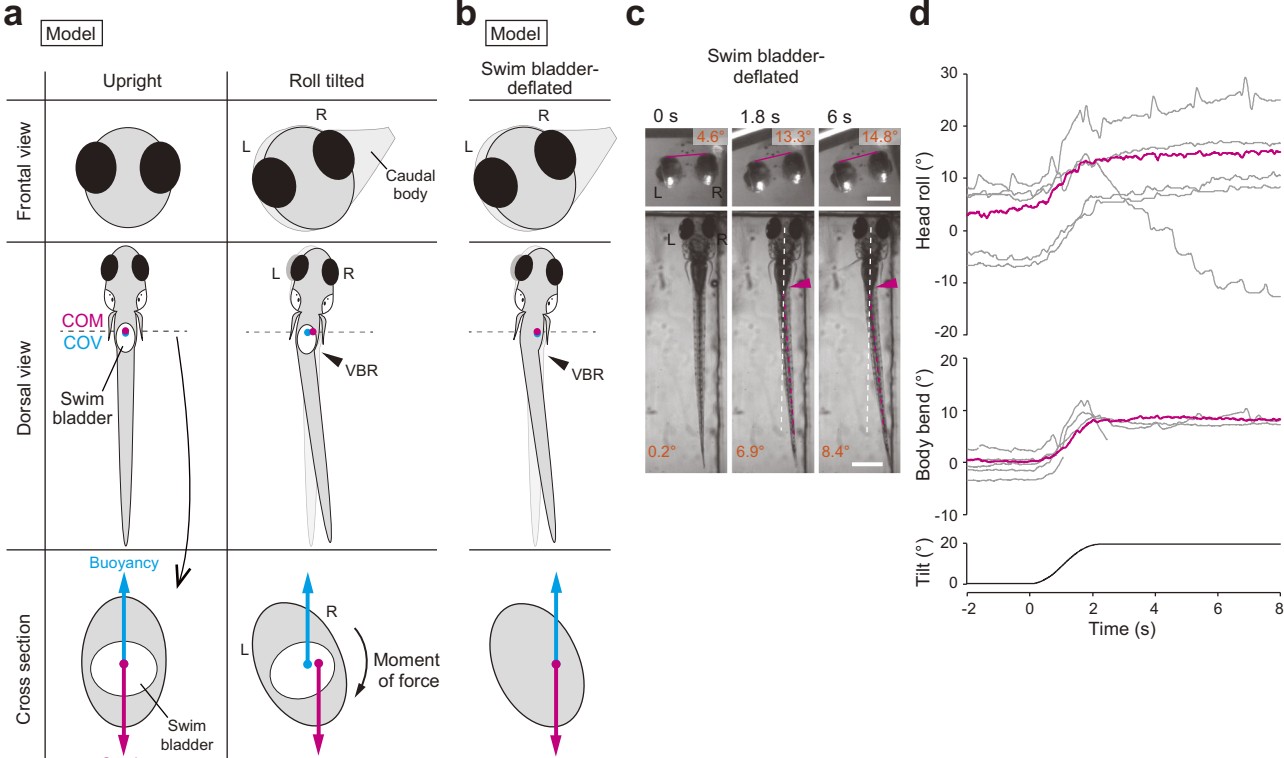

**Fig. 2 | Biomechanics for the VBR-mediated postural recovery. a** A model of postural recovery from roll tilt by the VBR. Frontal and dorsal views and cross-section around the swim bladder. Gravity and buoyancy act at the center of mass (COM) and center of volume (COV), respectively. The COM and COV are located near the swim bladder, and the COM is located slightly rostral to the COV[24]. In the dorso-ventral axis, the model assumes that the COM and COV are located in the same position. Left: the COM and COV are on the midline. Right: in the fish performing the VBR upon a roll tilt, the head and caudal body move toward the ear-up side, while the body around the swim bladder moves toward the ear-down side (top and middle). The COM and COV move toward the ear-up direction in the cross-section (bottom). The COM moves more laterally than the COV (see main text). This results in misalignment between gravity and buoyancy, generating a moment of force that counter-rotates the body. **b** A fish model with a deflated swim bladder. Positions of the COM and COV are the same even when fish perform the VBR (middle and bottom). Gravity and buoyancy are antiparallel on the same vertical axis (bottom). This does not generate a moment of force. **c, d** Behavioral experiments on fish with the swim bladder deflated. **c** Snapshots of the frontal and dorsal images of a fish during a left-down tilt. **d** Traces of the head roll and body bend angles of a fish in response to roll tilt. Six trials in one fish are shown. The magenta traces correspond to the trial shown in **c**. Scale bars, frontal images 200 μm; dorsal images 500 μm. Source data are provided as a Source Data file.

tangential nucleus (TAN), a vestibular nucleus involved in vestibulo-occular reflex[25], were activated during vestibular stimuli[18,19]. These studies also showed that neurons in the nucleus of medial longitudinal fasciculus (nMLF) were activated. Anatomical evidence showed that nMLF neurons receive axonal projections from neurons in the TAN[25]. A study reported that a VBR-like behavior occurred upon optogenetic activation of nMLF neurons[26]. The authors also provided evidence that a class of muscles, called posterior hypaxial muscles (PHMs), is involved in this bending behavior. Based on these previous studies, we hypothesized that the VBR is produced by vestibular inputs through the TAN–nMLF–PHM pathway (Fig. 3). To test this hypothesis, we performed Ca²⁺ imaging and cell-ablation experiments for each cell population in this pathway. The results are described in the sections below.

### TAN neurons are active during roll tilt when the ipsilateral ear is down

TAN neurons are located in the lateral-most region in the hindbrain rhombomere 5[16,25]. To genetically identify TAN neurons, we looked for transgenic fish lines expressing reporter genes in prospective TAN neurons and found that *evx2* transgenic fish met this criterion. In the image shown in Fig. 4a, prospective TAN neurons, which were identified by their position, were highlighted in a Tg(*evx2*:Gal4; UAS:Dendra2) fish[27,28] using a photoconversion technique. The axons of these neuronal population projected contralaterally and bifurcated to

ascend and descend along the medial longitudinal fasciculus. The ascending axons of TAN neurons extended to the midbrain, while the descending axons reached the third segment of the spinal cord (see also single-cell morphologies; Supplementary Fig. 2a, b). The axonal trajectory of these neurons was identical to that of TAN neurons reported in the previous studies[25,29]. Thus, we concluded that the labeled neurons in the *evx2* transgenic fish were TAN neurons. Crossing Tg(*evx2*:GFP) to RFP-expressing transgenic fish lines for neurotransmitter phenotypes revealed that GFP-positive TAN neurons were positive for RFP reporters for *vglut1* or *vglut2a* but not for *glyt2* and *gad1b* (Supplementary Fig. 2c), indicating that TAN neurons labeled in *evx2* transgenic fish are mostly excitatory.

We aimed to perform Ca²⁺ imaging in TAN neurons during roll tilts, using the imaging system in which an objective lens and a fish were tilted together using a motorized rotation stage[20] (Fig. 4b; Supplementary Fig. 2d). In this imaging system, two-color ratiometric imaging is used to reduce the motion-related artificial fluorescence intensity change during the tilt. Because this study is the first to use a wide-field version of the tiltable objective microscope (note that a confocal microscopy is used in the original version[20]), we evaluated the level of artifacts upon roll tilts using green/red-Dendra2 (the green Dendra2 was partially photoconverted to red-Dendra2 by ultraviolet light such that TAN neurons contained both green and red-Dendra2; Supplementary Fig. 2e). The amplitude of the green/red ratio change (ΔR/R₀) was confined within ±0.07 (Supplementary Fig. 2f). Thus,

$\Delta R/R_0$ that is above 0.07 was considered to be signals derived from neuronal activity.

We performed $Ca^{2+}$ imaging in TAN neurons, as a population, using larvae of the Tg(*evx2*:tdTomato-jGCaMP7b) transgenic fish that were generated in this study (jGCaMP7b is a green fluorescent $Ca^{2+}$ indicator, while tdTomato is a calcium-insensitive red fluorescent protein). In a representative example shown in Fig. 4c, a large increase in $\Delta R/R_0$ was observed in the left TAN neuron population when the fish was tilted in the left-down direction. Similarly, a large increase in $\Delta R/R_0$ was observed in the right population when the fish was tilted in the right-down direction. Similar phenomena were observed in other fish (seven fish; Fig. 4d). The maximum ratio changes (0.2–0.5) were much higher than the artificial changes, indicating that TAN neurons were active during the ipsilateral-down (ipsi-down) tilt. TAN neurons also exhibited slight increases in $\Delta R/R_0$ (0.1–0.2) during the ipsi-up tilt (Fig. 4c, d), but the maximum amplitude during the ipsi-down tilt was larger than that during the ipsi-up tilt (3.2-fold larger for the ipsi-down direction; Fig. 4d). In summary, the results indicate that TAN neurons are highly activated upon ipsi-down roll tilt.

## TAN neuron ablation impairs the VBR in the contralateral direction

Next, we examined the necessity of TAN neurons for producing the VBR. For this purpose, we unilaterally ablated TAN neurons using a laser (Fig. 5a; all the GFP-labeled TAN neurons, approximately 18 cells, were subjected to ablation) and examined the VBR performance during roll tilt in the head-embedded condition. When the left TAN neurons were laser ablated (Fig. 5a), the VBR to the right during the left-down tilt was greatly impaired, whereas that to the left during the right-down tilt was largely unaffected (Fig. 5b, c). The VBR performances during ablated-side-down tilts were greatly impaired, and this similar tendency was observed in seven fish (Fig. 5d). As a control ablation experiment, a similar number of *evx2*-positive neurons located ventromedial to the TAN neurons were ablated (Fig. 5e). In these animals, clear VBRs were observed during tilts in both directions (Fig. 5f–h).

## Optogenetic activation of TAN neurons induces the VBR in the contralateral side

In addition to the ablation experiments described above, we performed optogenetic activation experiments by expressing channelrhodopsin in TAN neurons using Tg(*evx2*:CoChR-GFP) fish. Optogenetic activation of TAN neurons elicited the VBR with a bend direction that was opposite to the illumination (Fig. 5i–k, o). In a control experiment, optogenetic activation of non-TAN neurons elicited the VBR negligibly (Fig. 5l–n, o). Collectively, the ablation and optogenetic activation experiments indicate that TAN neurons play a critical role in producing the VBR in the contralateral direction.

## nMLF neurons are active during ipsilateral-up (contralateral-down) roll tilt

nMLF neurons, a cluster of RS neurons that project axons into the spinal cord along the medial longitudinal fasciculus[30], were the next target of our investigation. We performed $Ca^{2+}$ imaging in nMLF

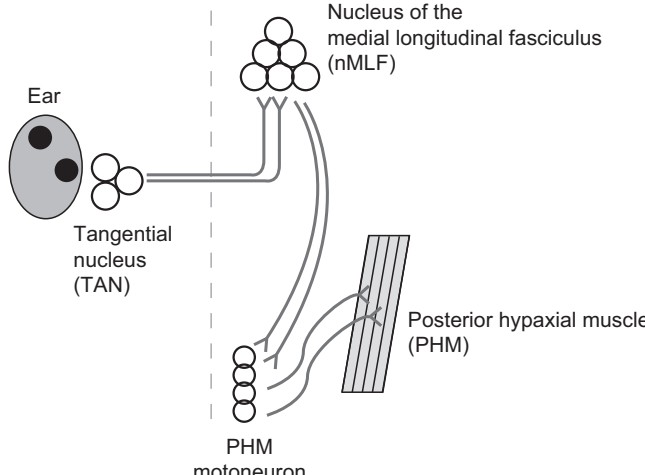

**Fig. 3 | A hypothesis of the neuronal circuits that produce the VBR.** Vestibular inputs activate neurons in the tangential nucleus (TAN). TAN neurons project to neurons in the nucleus of the medial longitudinal fasciculus (nMLF) on the contralateral side. The axons of the nMLF neurons descend to the spinal cord and project to motoneurons that innervate posterior hypaxial muscles (PHMs). Rostral is toward the top. The dashed line shows the midline.

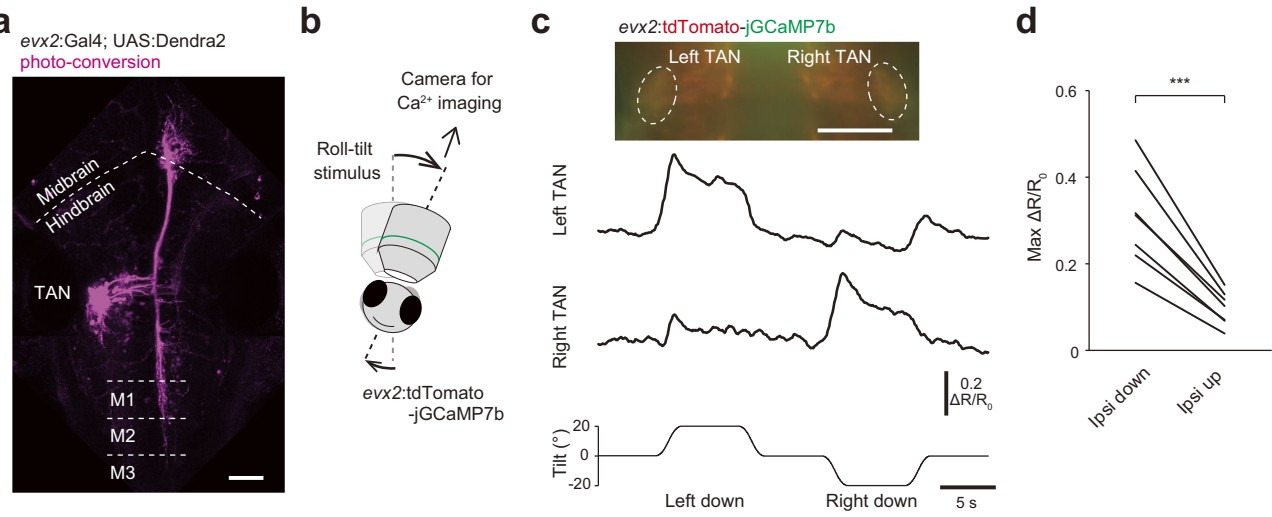

**Fig. 4 | $Ca^{2+}$ imaging of TAN neurons during roll tilts. a** TAN neurons photo-converted from green to red in Tg(*evx2*:Gal4; UAS:Dendra2) at 5 dpf. Confocal stacked image (maximum intensity projection). M1, M2, and M3 indicate the first, second, and third muscle segments, respectively. **b** Schematic of $Ca^{2+}$ imaging experiments. A fish embedded in agarose is imaged using a tiltable objective microscope. **c** Top: image of a Tg(*evx2*:tdTomato-jGCaMP7b) fish. Middle and bottom: time course of $\Delta R/R_0$ in the TAN neurons in response to a roll tilt. **d** Pairwise comparison of the maximum $\Delta R/R_0$ in TAN neurons between ipsi-down and ipsi-up tilts (seven fish). Four to six trials were performed in a fish, and the average values are shown for each fish. $p = 0.0003$ (two-sided paired samples $t$ test). Scale bars, 50 μm. Source data are provided as a Source Data file.

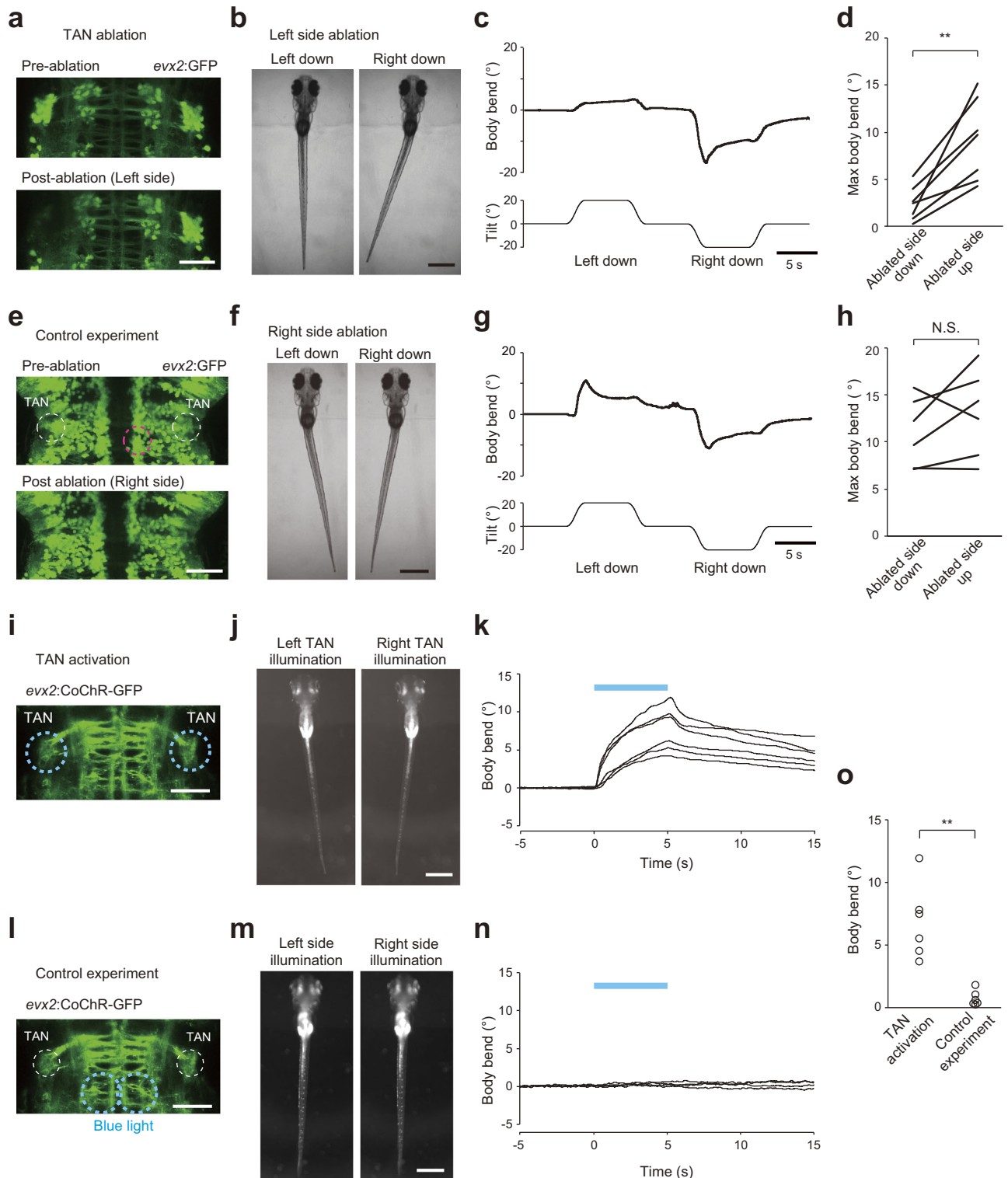

neurons at single-cell resolution using a tiltable objective microscope with a spinning-disk confocal unit[20]. For this purpose, nMLF neurons were retrogradely labeled with a green chemical fluorescent $Ca^{2+}$ indicator, Cal-520 dextran, and a red chemical fluorescent dye, rhodamine dextran. Figure 6a shows images with a maximum intensity projection and a single optical plane. During the tilt in the ipsi-up direction, a large increase in $\Delta R/R_0$ was observed in subsets of small-sized nMLF neurons (cell #1, #7, and #8 in Fig. 6b). Uniquely identifiable large nMLF neurons (MeLr, MeLc, MeLm), however, exhibited little or a relatively low level of $\Delta R/R_0$ increase during the tilt (cell #4,

#5, and #10 in Fig. 6b). Among all the nMLF neurons examined from ten fish ($n = 330$), approximately one-third of the neurons were obviously active when fish were tilted in the ipsi-up direction (red, orange, yellow, and white color in the pseudo-colored image in Fig. 6c). This conspicuous ipsi-up (contra-down) activity was consistent with our hypothesis, which is shown in Fig. 3 (note that nMLF neurons receive commissural axonal projections from the excitatory TAN neurons). During ipsi-down tilt, in contrast, slight decreases in $\Delta R/R_0$ occurred in many nMLF neurons (the dark blue color becomes more blackish in the pseudo-colored image in Fig. 6c). This suggests

**Fig. 5 | Ablation of TAN neurons impairs the VBR, while optogenetic activation of TAN neurons induces the VBR. a–d** Ablation of TAN neurons. **a** Confocal stacked images of Tg(*evx2*:GFP) fish before and after laser ablation of left TAN neurons. **b** Behaviors of head-embedded fish during roll tilts. **c** Time course of the body bend angle upon roll tilt. The same trial shown in **b**. **d** Pairwise comparison of the maximum body bend angles between ablated-side-down and ablated-side-up tilts (three fish for left side ablation and four fish for right side ablation). Average values of two to four trials are shown for each fish. $p = 0.003$ (two-sided paired samples *t*-test). **e–h** Control experiment. **e** Confocal stacked images of Tg(*evx2*:GFP) fish before and after laser ablation of non-TAN neurons on the right side (magenta circle). **f** Same as **b** upon a control experiment. **g** Same as **c** upon a control experiment. The same trial shown in **f**. **h** Same as **d** upon a control experiment (two fish for left side ablation and four fish for right side ablation).

Average values of four to six trials are shown for each fish. $p = 0.23$ (two-sided paired samples *t* test). **i–k** Optogenetic activation of TAN neurons. **i** Confocal stacked image of Tg(*evx2*:CoChR-GFP). The blue dotted circles are areas illuminated with blue light. **j** Behaviors of head-embedded fish upon blue light illumination. **k** Time course of the body bend angles. Positive values indicate body bend to the contralateral direction with respect to the illumination. The blue bar indicates illumination that lasts for 5 s. Six trials in a fish are shown. **l–n** Control experiment. **l** Same as **i**, but different areas (blue dotted circles) were illuminated as a control experiment. **m** Same as **j** upon a control experiment. **n** Same as **k** upon a control experiment. Four trials in a fish are shown. **o** Body-bend angle upon optogenetic activation (mean angle between 4 and 5 s). Six fish for each. $p = 0.004$ (two-sided two-sample *t*-test). Scale bars, **a, e, i, l** 50 μm; **b, f, j, m** 500 μm. Source data are provided as a Source Data file.

that these neurons were slightly inhibited during ipsi-down tilt. To quantify what percentage of neurons were active, we set the threshold to 0.1 for a $\Delta R/R_0$ increase, and neurons with a maximum $\Delta R/R_0$ that was above 0.1 were considered to be active based on the artificial fluorescent intensity changes during the tilt[20]. Based on this criterion, 41% of neurons (135/330) were active only during the ipsi-up tilt, 15% (48/330) were active during both the ipsi-up and -down tilts, and a small proportion of neurons (8/330, 2%) was active only during the ipsi-down tilt (Supplementary Fig. 3a). Therefore, during the ipsi-up tilt, more than half (55%) of the nMLF neurons were judged to be active.

We further characterized active neurons with respect to their size and position. For the soma size, highly active neurons (e.g., those neurons with a maximum $\Delta R/R_0$ above 0.5) were mostly small with a soma area size that was less than 60 μm² (Fig. 6d). As noted above, uniquely identifiable large nMLF neurons (MeLr, MeLc, MeLm, and MeM) exhibited little or a relatively low-level of activity. To investigate the relationship between neuronal activity and soma position, we aligned the imaged neurons in a horizontal plane and in the dorso-ventral axis (Supplementary Fig. 3b, e; see Methods). For the rostro-caudal and medio-lateral axes, no clear relationship was found between the maximum $\Delta R/R_0$ and soma positions (Supplementary Fig. 3c, d). For the dorso-ventral axis, there was a tendency that neurons located dorsally exhibited a higher maximum $\Delta R/R_0$ (Supplementary Fig. 3e).

In summary, Ca²⁺ imaging revealed that a subset of nMLF neurons on the ear-up side was active during the tilt. Among nMLF neurons, highly active neurons were mostly small neurons, and they tended to be dorsally located.

### Ablation of nMLF neurons impairs the VBR to the ipsilateral direction

For laser ablation of nMLF neurons, we sought to genetically label them. Previous studies reported that nMLF neurons are labeled in *pitx2* transgenic fish[31]. We first examined the percentage of labeled nMLF neurons in Tg(*pitx2*:Dendra2) (generated in this study) by performing retrograde labeling with rhodamine dye into the larvae of the transgenic fish, and we found that approximately 80% of backfilled nMLF neurons, including small neurons, were positive for Dendra2 (Supplementary Fig. 4). In Tg(*pitx2*:Dendra2) transgenic fish, Dendra2 expression was not limited to nMLF neurons in the brain (Supplementary Fig. 4). To specifically visualize nMLF neurons, we highlighted nMLF neurons using an optical backfill technique[32]; Fig. 7a). Then, we unilaterally ablated photoconverted nMLF neurons (all the optically backfilled nMLF neurons, approximately 25 cells, were subjected to laser ablation; Fig. 7b), and the resultant larvae were examined for their VBR performance upon roll tilts.

A representative example is shown in Fig. 7c, d. The larva (left side ablation) exhibited a clear impairment in the leftward VBR during left-up tilt, whereas the rightward VBR during right-up tilt was largely unaffected. The VBR performances during ablated-side-up tilts were greatly impaired, and this similar tendency was observed in eight fish (Fig. 7e). These results (impairments of the VBR to the ablated side) are consistent

with our hypothesis shown in Fig. 3. As a control ablation experiment, a similar number of Dendra2-positive neurons located in the hindbrain were ablated (Fig. 7f). In these animals, clear VBRs were observed during tilts in both directions (Fig. 7g–i). Thus, the ablation experiments indicate that nMLF neurons play an important role for producing the VBR in the ipsilateral direction during the ipsi-up head tilts.

### Slow-type PHMs are active during ipsilateral-up (contralateral-down) roll tilt

PHMs, a class of hypaxial muscles that run obliquely near the swim bladder[33], were the last target of our investigation. We first performed anatomical investigations to investigate whether PHMs contained both fast- and slow-type muscles. Images of compound transgenic fish of Tg(*α-actin*:GFP)[34] (a marker for fast-type muscles) and Tg(*smyhc2*:loxP-RFP-loxP-DTA) (generated in this study; a marker for a subset of slow-type muscles) revealed that PHMs consisted of both fast- and slow-type muscles with thin slow-type muscles intercalating with thick fast-type muscles (Supplementary Fig. 5a, b). Images also showed that PHMs were divided into three segments (Supplementary Fig. 5a), as previously reported[35,36]. We defined these segments as the rostral, middle, and caudal segments.

We first focused on slow-type PHMs. The PHMs were imaged from the ventral side with the wide-field version of the tiltable objective microscope (Supplementary Fig. 2d). In Ca²⁺ imaging of the PHMs, artificial fluorescent changes were derived not only from the tilt of a sample (Supplementary Fig. 2e, f), but also from muscle displacement during the VBR. To estimate the level of artifacts in this experiment, we used Tg(*pitx2*:Dendra2) fish that express green/red-Dendra2 in slow-type PHMs (Supplementary Fig. 5d, e). During tilt, the maximum $\Delta R/R_0$ was mostly confined within 0.1 (Supplementary Fig. 5f, g).

Next, we performed Ca²⁺ imaging in slow-type PHMs using Tg(*smyhc2*:tdTomato-jGCaMP7b). Simultaneously, behaviors of the head-restrained fish were observed from the dorsal side (Fig. 8a and Supplementary Fig. 5c; see Methods). Fluorescence derived from bilaterally located slow-type PHMs was clearly visible (Fig. 8b). During the imaging experiments, we noticed that slow-type PHMs occasionally exhibited bursts of rhythmic spontaneous activity (Supplementary Fig. 5h; the physiological significance of this activity is unknown). To simplify the analysis, we collected data from the trials in which the bursts of spontaneous activities did not occur. Figure 8c shows an example of $\Delta R/R_0$ from the slow-type PHMs in each segment during roll tilts. Upon a right-up tilt, large increases of $\Delta R/R_0$ were observed in the middle and caudal segments on the right side. Conversely, upon a left-up tilt, large increases of $\Delta R/R_0$ were observed in the middle and caudal segments on the left side. For the rostral segment, increases in $\Delta R/R_0$ upon roll tilts were present, but their amplitude was not large compared with those of the middle and caudal segments. Quantitative analyses of population data collected from six fish are shown in Fig. 8d. All three segments occasionally exhibited small $\Delta R/R_0$ increases that were considered active during the ipsi-down tilt, but mean amplitudes of the maximum $\Delta R/R_0$ during the ipsi-up tilt were larger than those

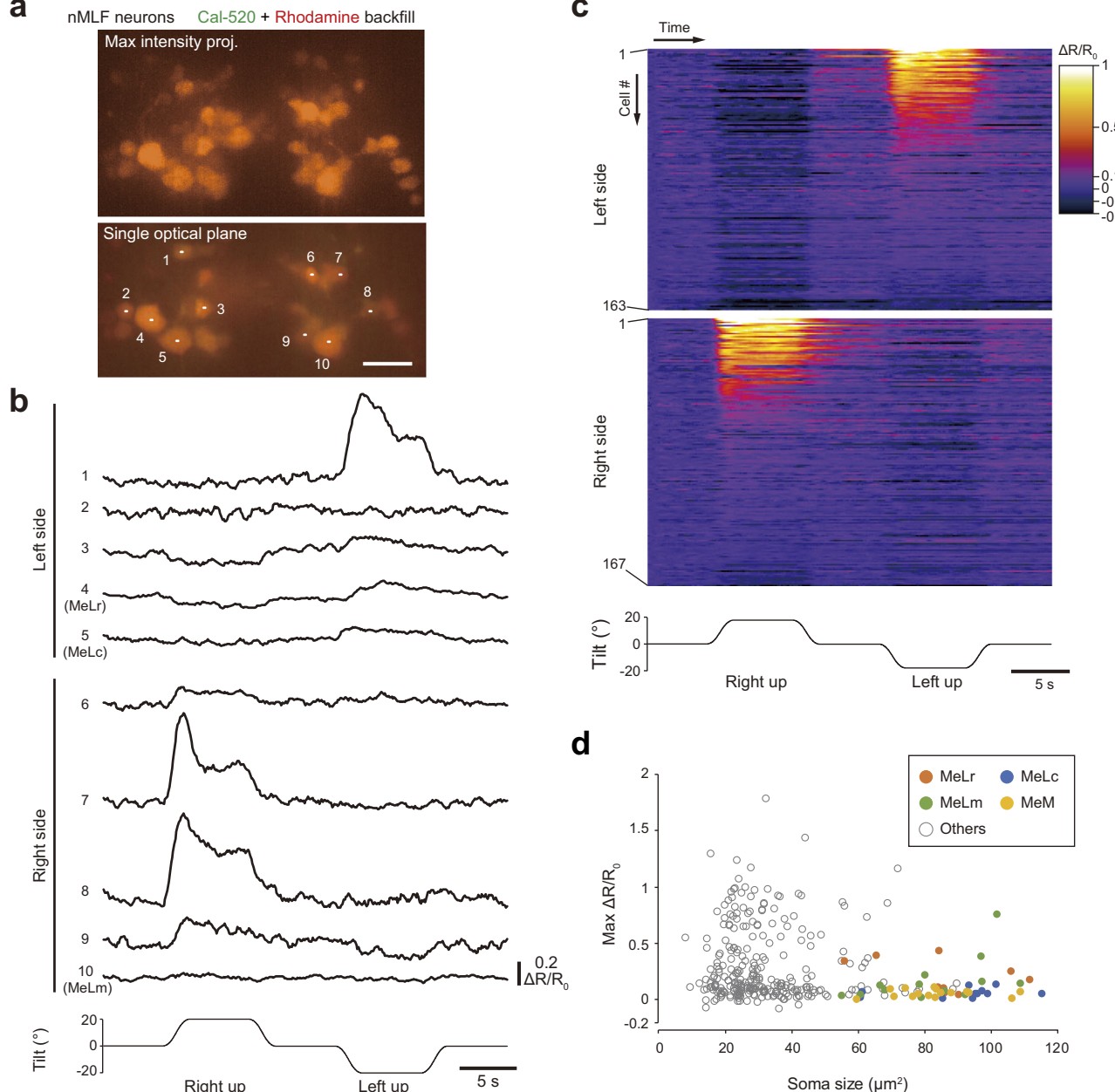

**Fig. 6 | Ca²⁺ imaging of nMLF neurons during roll tilts. a** Top: confocal stacked image of nMLF neurons that were retrogradely labeled with Ca²⁺ indicator, Cal-520, and rhodamine dextran. Bottom: single optical section. Scale bar, 20 μm. **b** Time course of ΔR/R₀ in each neuron in response to roll tilt. The numbers correspond to those in **a**. **c** Color-coded ΔR/R₀ traces of all nMLF neurons (330 neurons from 10 fish, with 163 for the left side and 167 for the right side) in response to a roll tilt. Neurons are grouped by their location (left or right) and are sorted by the maximum ΔR/R₀ during ipsi-up tilt. Neuronal indexes are denoted on the left. **d** Graph showing the maximum ΔR/R₀ during ipsi-up tilt (*Y* axis) vs. the soma area size (*X* axis). Source data are provided as a Source Data file.

during the ipsi-down tilt (2.0-fold for the rostral, 5.7-fold for the middle, and 3.5-fold for the caudal segments). Concomitant with the large increase of ΔR/R₀ in slow-type PHMs, the body bent to the side where the PHMs were highly recruited during the roll tilts (Fig. 8c).

We next examined activities of fast-type PHMs. For this purpose, we generated Tg(*α-actin*:tdTomato-jGCaMP7f) (Supplementary Fig. 6a). Ca²⁺ imaging in fast-type PHMs revealed no significant increase in ΔR/R₀ in the three segments during the tilt (Supplementary Fig. 6b, c). Collectively, the results of Ca²⁺ imaging indicate that slow-type PHMs, not fast-type PHMs, located in the ear-up side are active when fish perform the VBR. Among slow-type PHMs, those in the middle and caudal segments exhibited higher activity than those in the rostral segment.

## Slow-type PHMs play a critical role for producing the VBR

Lastly, we performed genetic ablation experiments on slow-type PHMs. For this purpose, we generated Tg(*tbx2b*:Cre) transgenic fish (*tbx2b* is expressed in migratory muscle precursors including muscle precursors for PHMs[36]), and crossed the fish with Tg(*smyhc2*:loxP-RFP-loxP-DTA) fish. In the resultant compound transgenic fish, Cre-mediated recombination occurred in PHMs, as shown by the absence of RFP in PHMs (Fig. 9a, left panels). This led to the expression of diphtheria toxin A fragment (DTA), which would kill slow-type PHMs. Successful ablation of slow-type PHMs was examined by immunostaining with the S58 antibody. In Cre-negative control sibling fish, slow-type muscles in both the trunk muscles and PHMs were labeled, whereas in the compound transgenic fish, labeling in PHMs

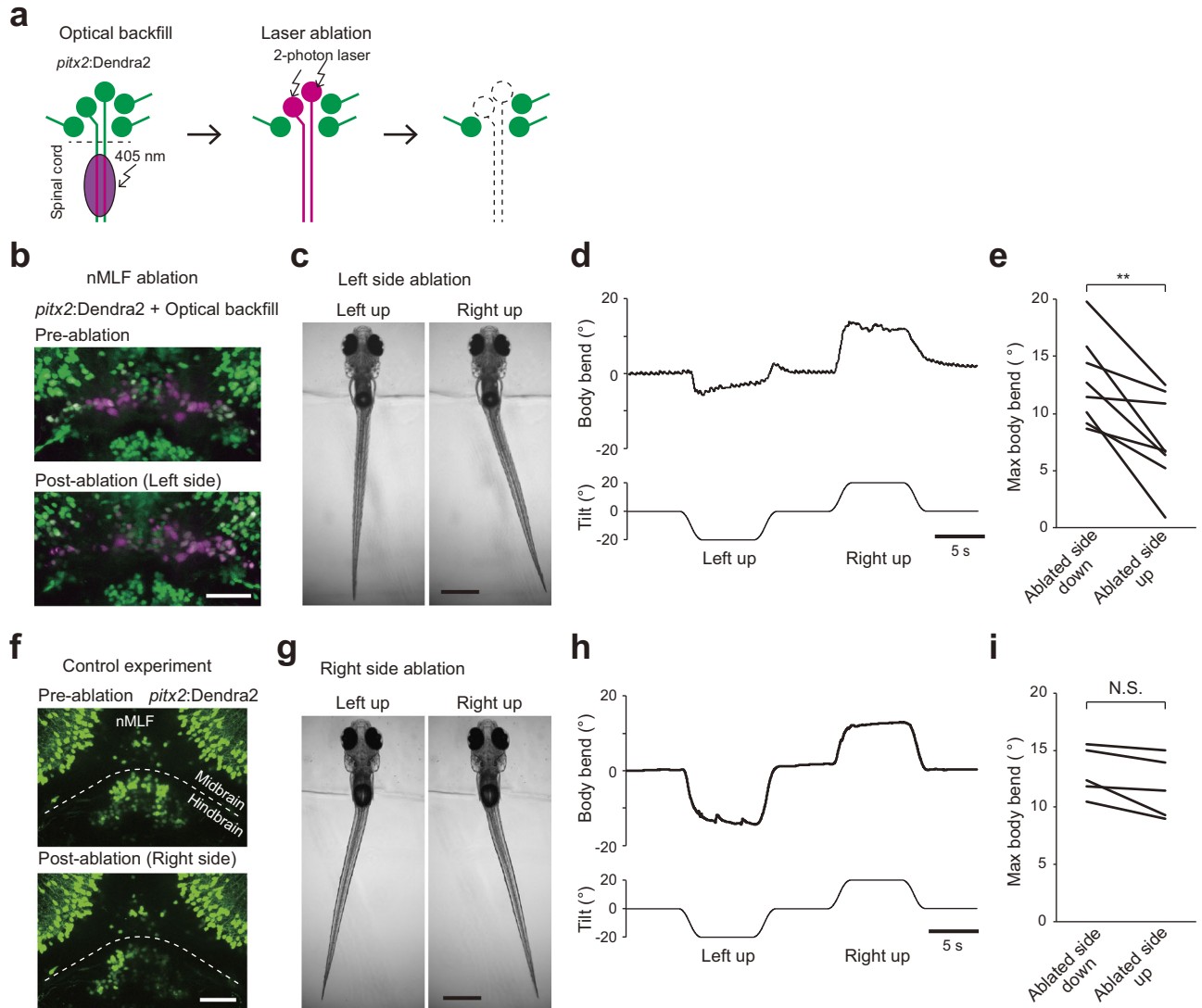

**Fig. 7 | Ablation of nMLF neurons impairs the VBR. a–e** Ablation of nMLF neurons. **a** Schematic showing optical backfill followed by laser ablation of nMLF neurons. **b** Confocal stacked images before (top) and after (bottom) laser ablation of nMLF neurons on the left side. **c** Behaviors of head-embedded fish during roll tilts. **d** Time course of the body bend angle in response to roll tilt. The same trial shown in **c**. **e** Pairwise comparison of maximum body bend angles between ablated-side-down and ablated-side-up tilts (eight fish; four for each side ablation). Average values of at least five trials are shown for each fish. $p = 0.003$ (two-sided paired samples $t$ test). **f–i** Control experiment. **f** Confocal stacked images fish before (top) and after (bottom) laser ablation of hindbrain neurons (non-nMLF neurons) in Tg(*pitx2*:Dendra2). **g** Same as **c** upon a control experiment. **h** Same as **d** upon a control experiment. The same trial shown in **g**. **i** Same as **e** upon a control experiment (five fish, with three for left side ablation and two for right side ablation). Average values of five to eight trials are shown for each fish. $p = 0.05$ (two-sided paired samples $t$-test). Scale bars, **b**, **f** 50 μm; **c**, **g** 500 μm. Source data are provided as a Source Data file.

was almost completely absent while labeling in the trunk muscles was intact (Fig. 9a, middle and right panels).

We examined the VBR performance of the compound transgenic fish during roll tilt in the head-embedded condition. As shown in Fig. 9b, c, VBRs of these fish were markedly reduced. Quantitative analyses of the population data verified that the maximum body bend angles in the compound transgenic fish were three-times smaller than those in Cre-negative control siblings (six fish for each; Fig. 9d). These results indicate that slow-type PHMs play a critical role for producing the VBR during the roll tilts.

## Discussion

Fish possess a capability to recover from a roll-tilted posture by performing swimming[21,22]. However, it remains unclear whether fish possess a postural control mechanism without swimming. An artificial otolith displacement was shown to induce a body bend

reflex that does not accompany swimming[23], but the physiological role of this behavior was unknown. In this study, we demonstrated that the body bend reflex, which we called VBR, is crucial for the postural recovery from a roll tilt without swimming. We also provided a physical model that explains how the VBR recovers upright posture. When a roll tilt occurs, the fish performs the VBR. This bend deflects the head and caudal body toward the ear-up side, and in reaction, the body around the swim bladder moves toward the ear-down side. Because the swim bladder has a low-density, the VBR unequally shifts the COM and COV. This results in a misalignment between gravity and buoyancy, generating a moment of force with a rotational direction toward recovering the upright posture (Fig. 2a). This model was strongly supported by experiments in which the swim bladder was deflated (Fig. 2b–d). Without gas in the swim bladder, the above-mentioned unequal shifts of the COM and COV would not occur. Fish with deflated swim

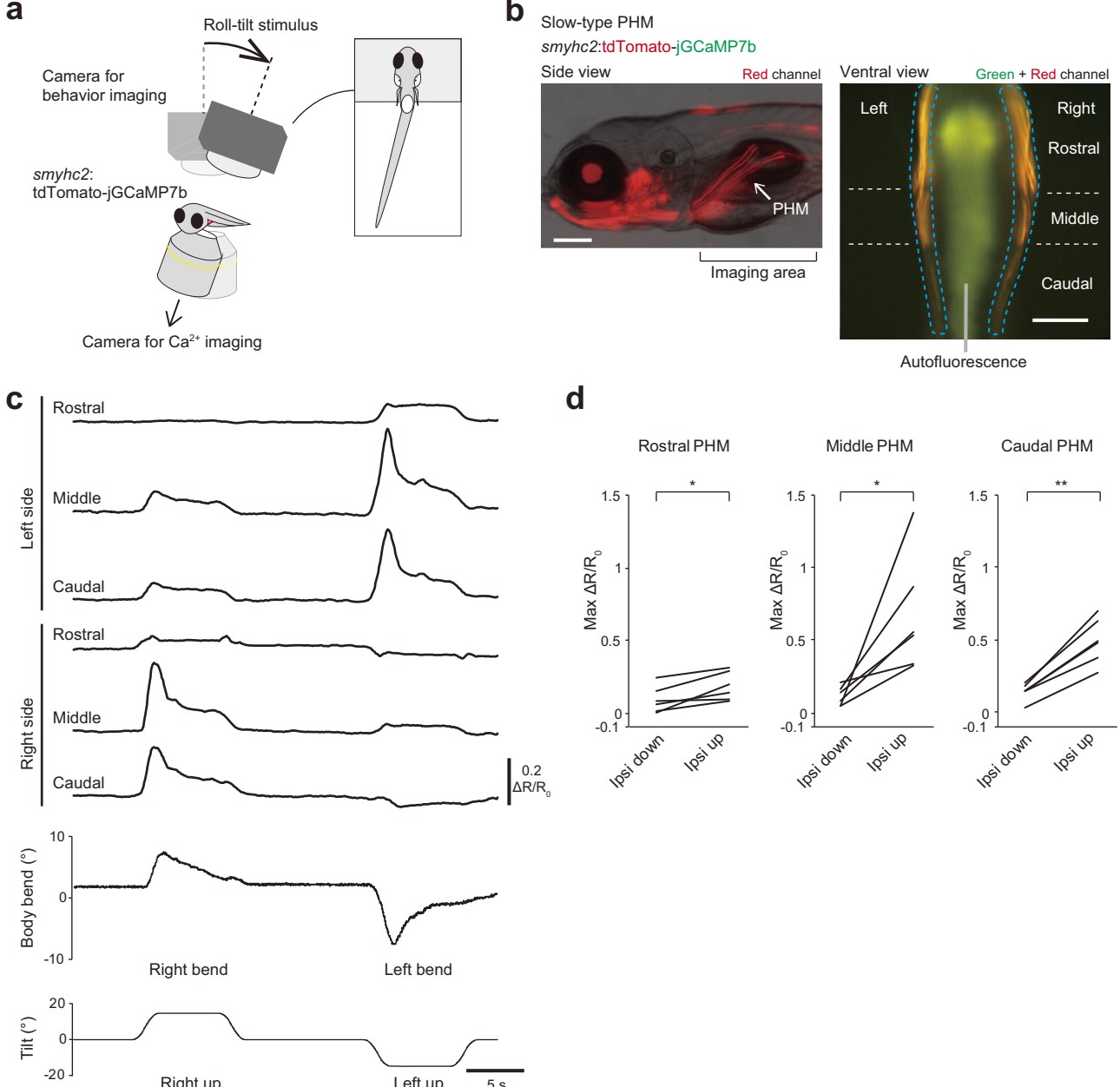

**Fig. 8 | Ca²⁺ imaging of slow-type PHMs during roll tilts. a** Schematic showing simultaneous imaging of Ca²⁺ in PHMs and fish behaviors. **b** Fluorescence images of Tg(*smyhc2*:tdTomato-jGCaMP7b) fish at 6 dpf. Left: lateral view. Red fluorescence and transmitted light images are merged. Right: ventral view of the area around PHMs ("imaging area" in the left panel). Rostral is to the top. Green and red channels are merged. PHMs are bilaterally located (dashed blue lines). They consist of the following three segments: rostral, middle and caudal segments. Autofluorescence (greenish signal) derived from intestine and yolk is present in the middle. Scale bars, 200 μm. **c** Time course of ΔR/R₀ in each segment of slow-type PHMs and body bend angles in response to a roll tilt. **d** Pairwise comparison of maximum ΔR/R₀ in each segment of slow-type PHMs between ipsi-down and ipsi-up tilts (six fish). A single trial or average values of two trials are shown for each fish. $p = 0.01$ for the rostral, $p = 0.02$ for the middle, and $p = 0.0006$ for the caudal segments (two-sided paired samples $t$ test). Source data are provided as a Source Data file.

bladders were unable to recover from a tilted posture although they continued to perform the VBR.

Taking physics into account, the absolute location of the COM in space does not change by the body bend with the internal muscular contractions (Supplementary Fig. 1k, l; note that the apparent shift of the COM in the cross-sectional view in Fig. 2a is caused by the lateral movement of the body around the COM). From this view, the VBR can be considered to be a behavior to move the low-density part of the body (i.e. swim bladder) toward the ear-down side, resulting in the lateral shift of the COV (Supplementary Fig. 1l). The strength of the recovery moment of force depends on the separation distance between the COM and COV. The maximum separation distance (and thus, the maximum moment of force), in theory, can be obtained when the following two conditions are met: (1) the swim bladder, which is the least dense part of the body, is located near the COM; and (2) the body bend occurs near the swim bladder. Indeed, the fish body shape and the bend location fulfill these conditions, as follows: (1) the COM is located near the swim bladder[24]; and (2) the bend occurs near the swim bladder (Fig. 1d). Our calcium imaging experiments in slow-type PHMs provide further evidence for the bend location. The most conspicuous activities were observed in the middle and caudal segments of PHMs (Fig. 8c). These two segments are located directly lateral or latero-

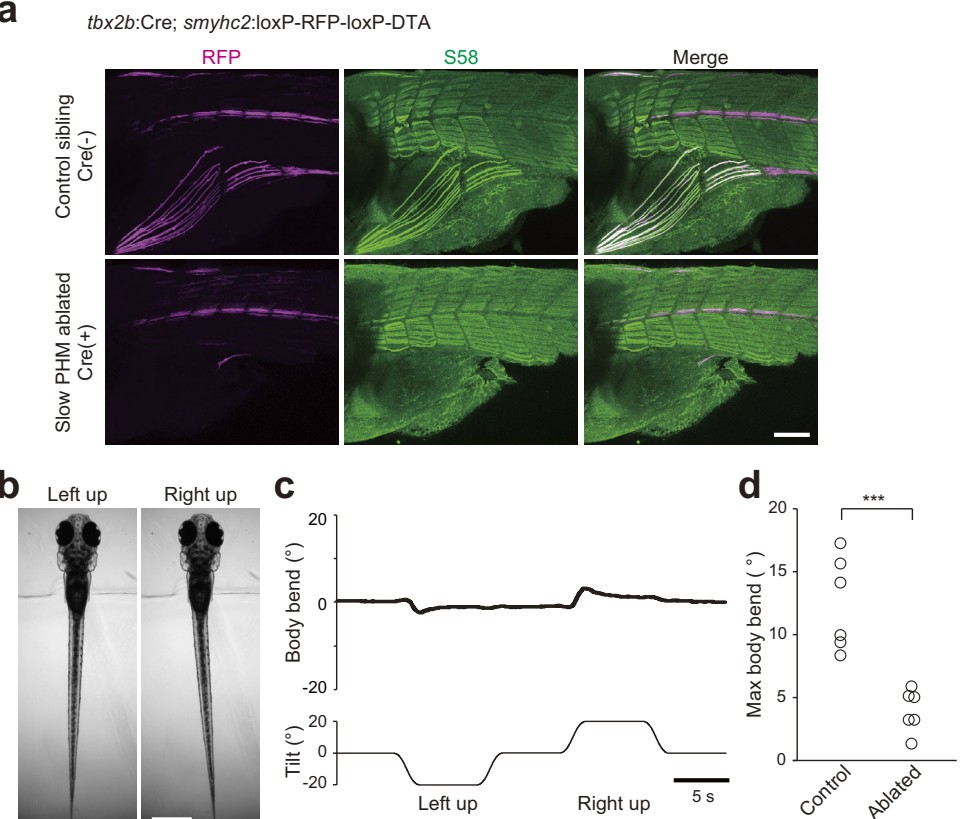

**Fig. 9 | Ablation of slow-type PHMs impairs the VBR. a** Images of immunohistochemistry with S58 antibody against Tg(*smyhc2*:loxP-RFP-loxP-DTA) fish. Top: images of a fish without *tbx2b*:Cre (control). Bottom: images of a fish with *tbx2b*:Cre. **b** Behaviors of Tg(*tbx2b*:Cre; *smyhc2*:loxP-RFP-loxP-DTA) fish in the head-embedded condition during roll tilts. **c** Time course of the body bend angle in response to roll tilt. The same trial as shown in **b**. **d** Maximum body bend angles of control (without Cre) and PHM-ablated (with Cre) fish (six fish for each). Average values of at least five trials are shown for each fish. $p = 0.0004$ (two-sided two-sample $t$ test). Scale bars, **a** 100 μm; **b** 500 μm. Source data are provided as a Source Data file.

caudal to the swim bladder (Supplementary Fig. 5a). Thus, the VBR is a highly reasonable behavior when physics is taken into account, and it is likely to be an energy-efficient behavior for postural correction.

Terrestrial animals are unstable when standing upright, and they maintain their upright posture by continuously performing fine, static postural control[2,3,37]. In larval zebrafish, the vertical positions of the COM and COV are close to each other[24], suggesting that the zebrafish is also unstable in the roll direction. Based on the results of the present study, we suggest that zebrafish keep their dorsal-up posture by frequently performing the VBR upon small disturbances. If so, the postural control by the VBR in fish may correspond to the static control in land-walking vertebrates.

Mechanically, postural recovery of larval zebrafish using the VBR shows similarities with ship stability through use of a misalignment between gravity and buoyancy. However, there is a fundamental difference between the two systems, which is that the ship stability is automatically generated, while the VBR is actively generated with the neural and muscular activities. Theoretically, fish can acquire stability in the roll axis if the COV is located far above the COM (i.e., placing the swim bladder in an extreme dorsal region of the body). Why do larval zebrafish (and perhaps other fish) not adopt this stable design? One possibility is that fish select maneuverability over stability[38]. Another possibility is that the swim bladder cannot be situated in such an extreme dorsal location owing to the necessity to take up gas from the gastrointestinal tract. In any case, animals are likely to efficiently compensate for instability through use of the fewest movements (i.e., VBR) to achieve stability.

The vestibulo-spinal reflex involves vestibulo-spinal (VS) neurons that project directly to the spinal cord[6–8]. However, the vestibulo-spinal reflex is thought to be controlled not only by direct pathways through VS neurons, but also by indirect pathways through vestibular nuclei and RS neurons[9]. In lamprey and mammals, RS neurons respond to vestibular stimuli[39–42]. Recently, the functional importance of RS neurons for the vestibulo-spinal reflex was shown[13]. Our results clearly showed that circuits from neurons in the vestibular nucleus (TAN neurons) to RS neurons (nMLF neurons) play a critical role in the vestibulo-spinal reflex in zebrafish. This study supports the evolutionary conservation and importance of the indirect pathways in vertebrate vestibulo-spinal reflexes.

TAN neurons project to the extraocular motoneurons (nIII and nIV) and are involved in the vestibulo-ocular reflex[25]. Anatomical studies strongly suggested that, in addition to the nIII and nIV, TAN neurons also directly project to the nMLF neurons[25,29]. Results of the present study are consistent with this neuronal projection and further revealed that the TAN−nMLF pathway is involved in a postural control. Thus, TAN neurons have dual functions because they play roles in both vestibulo-ocular and vestibulo-spinal reflexes. In mammals, a class of vestibular neurons projects directly to both the extraocular and spinal motoneurons[43,44]. Thus, the presence of vestibular neurons that are involved in both vestibulo-ocular and vestibulo-spinal reflexes is likely conserved among vertebrate species. Based on the finding of the present study, we expect that in mammals, there are vestibular neurons that control vestibulo-ocular reflex directly and vestibulo-spinal reflex indirectly via RS neurons.

In the nMLF neuron ablation experiments, the body bend was not completely abolished. A likely explanation for this is that not all nMLF neurons were ablated, and the residual nMLF neurons contributed to producing the weakened VBR. Another possibility is that nMLF-independent pathways may be involved in the VBR. A likely candidate for this is the descending axons of TAN neurons. Our anatomical study revealed that the descending axons project to the rostral spinal cord, but do not appear to reach the somata of PHM-motoneurons (Fig. 4a, Supplementary Fig. 6d). Thus, even if the descending axons contribute to producing the VBR, they would exert the action via a polysynaptic pathway.

nMLF neurons are involved in locomotor behaviors[26,45–47]. A study showed that an optogenetic activation of nMLF neurons elicited a VBR-like bending behavior[26]. Because the study was performed in the context of examining functional roles of nMLF neurons during swimming, the authors suggested that the bending behavior would play a role in inducing biased swimming to the left or right, thereby enabling fish to change their swimming direction. While this may be a role of bending, our results showed that, independent of swimming, bending has a clear physiological function, which is postural correction in the roll axis.

Because the tilt stimuli occasionally elicit swimming, it is possible that nMLF neuron responses in our experiments may have been associated with the swimming and not associated with the VBR alone. While this possibility cannot be completely ruled out, the observed responses of nMLF neurons were highly asymmetric with almost no activation in the ear-down side (Fig. 6c), which strongly suggests that possible contributions of swimming-associated activities are minimal (note that the swimming-related activities of nMLF neurons occur bilaterally[26,45]). Furthermore, in our experiments, trials in which there were large displacements of the imaged field (which indicates strong movements by the fish) were excluded from the analyses. This probably contributed to reducing the chance of analyzing swimming-related neuronal activities.

Among nMLF neurons, highly active neurons were mostly small (Fig. 6d). This is consistent with the previous study showing that neurons with smaller somata had a greater impact on VBR-like behavior than those with larger somata[26]. nMLF neurons with smaller somata project to the rostral spinal cord[26], where PHM-motoneurons are located (Supplementary Fig. 6d). Thus, we expect that these small nMLF neurons make monosynaptic connections onto slow-type PHM-motoneurons, although this should be tested using paired electrophysiological recordings.

In this study, we showed that the TAN–nMLF pathway plays a critical role in fine postural control in the roll axis in larval zebrafish. However, TAN–nMLF is not the only pathway that conveys vestibular information to the spinal cord. VS neurons, which have axons that descend along the ipsilateral side of the spinal cord, have long been considered to be a main pathway in postural control[48,49], although little is known about their mechanisms of action due to the paucity of knowledge about their output connectivity. Does the VS pathway play a role in postural control in the roll axis? If so, in what situation is it involved? One attractive possibility is that the VS pathway is involved in the postural control during dynamic movements, i.e., swimming. If this is the case, investigating how fish use the two pathways (the ipsilaterally projecting VS pathway and the contralaterally projecting TAN–nMLF pathway) properly or cooperatively, depending on the type of behavior, would be an interesting area of future research.

Similar to fish, many vertebrate species use both fine (static) and vigorous (dynamic) control strategies for correcting posture[1–3,50,51]. The mechanism of how switching between the two strategies (fine versus vigorous) is achieved is an interesting question, and zebrafish could serve as a good model system to address this issue.

## Methods

### Animals

All procedures were performed in accordance with the guidelines approved by the animal care and use committees at the National Institute of Natural Sciences. Zebrafish adults, embryos, and larvae were maintained at 28.5 °C. All animals were kept on a 14:10 or 12:12 hour light:dark cycle, except for embryos and larvae expressing channelrhodopsin or Dendra2 proteins, which were kept under dark conditions. Morphological and immunostaining experiments were performed in 5 days post fertilization (dpf) larvae. Behavioral and imaging experiments were performed using 6 dpf larvae. Sex is not yet determined at the larval stage. At 5 dpf, larvae started to be fed.

The following previously published transgenic lines were used: Tg(*evx2*:GFP)[52], Tg(*evx2*:Gal4)[27], Tg(UAS:Dendra2)[28], Tg(*vglut2a*:RFP)[53], Tg(*glyt2*:RFP)[54], Tg(*gad1b*:RFP)[55], Tg(*α-actin*:GFP)[34], Tg(*vachta*:Gal4)[28], and Tg(UAS:Kaede)[56]. Additionally, Tg(*evx2*:tdTomato-jGCaMP7b), Tg(*evx2*:CoChR-GFP), Tg(*vglut1*:RFP), Tg(*pitx2*:Dendra2), Tg(*smyhc2*:tdTomato-jGCaMP7b), Tg(*smyhc2*:loxP-DsRed-loxP-DTA), and Tg(*tbx2b*:Cre) were generated using the CRISPR/Cas9-mediated knock-in method with the hsp70 promoter[27]. The sgRNA sequences for targeting the genes were as follows: ggagggagagccagaacaga (for *evx2*)[27], gagagagagactcgggcgcgcg (for *vglut1*), gagctttgactgtcagcgcg (for *pitx2*), gacttggatttcatctggcg or cacaatgctgcaagctcac (for *smyhc2*: the former was used for the generation of Tg(*smyhc2*:tdTomato-jGCaMP7b) while the latter was used for the generation of Tg(*smyhc2*:hs:loxP-RFP-loxP-DTA)), and ataaagcgtaagccgaccg (for *tbx2b*). Tg(*evx2*:CoChR-GFP) was generated using CoChR-GFP-Kv2.1 sequence[57]. Tg(*α-actin*:tdTomato-jGCaMP7f) was generated using the Tol2-mediated transgenesis[58] with *α-actin* promotor[34]. Tg(*smyhc2*:tdTomato-jGCaMP7b) and Tg(*α-actin*:tdTomato-jGCaMP7f) were generated with tdTomato-jGCaMP7b or tdTomato-jGCaMP7f fusion construct[59]. Zebrafish lines generated in this study have been deposited to the National BioResource Project in Japan.

### Behavioral experiments without head restraint

Behavioral experiments were performed using the device shown in Fig. 1a and Supplementary Fig. 1a. A larval fish at 6 dpf was transferred in a small acrylic chamber (1.2 [length] × 50 [width] × 1 [height] in mm). The chamber was then filled with fish-rearing water, and the top and the front sides were covered with cover slips. Then, the chamber was placed at the intersection of optical paths in a T-shaped unit attached to a motorized rotation stage (Thorlabs, HDR50/M). To take images from both the dorsal and frontal sides of the fish, two pairs of lenses (Olympus, ×5/NA 0.15) were placed on either side. In each lens pair, two lenses were oriented in opposite directions. As a light source, infrared light (Mightex; SLS-0208-E) was positioned ventral to the fish. The dorsal camera (Teledyne FLIR, GS3-U3-23S6M) and frontal camera (Basler, acA640-750um) were placed on the image planes. Tilt stimulus (speed: 15°/s, acceleration, and deceleration: 15°/s²) up to 20° was applied to the left- or right-down directions. Images were taken at 100 frames per second (fps) for both the frontal and dorsal cameras. The two cameras roll-tilted together with the chamber (Fig. 1a, Supplementary Fig. 1a). This means that the dorsal view of the fish became oblique, not vertical, when the fish performed posture-correcting behaviors.

In the experiments of Fig. 1d, e, the fish was immersed in 0.8% methylcellulose solution dissolved in fish-rearing water. For experiments with fin-amputated fish (Supplementary Fig. 1b, c), the fish were temporally anesthetized with 0.02% ethyl 3-aminobenzoate methanesulfonate (MS-222), and the pectoral fins were manually removed using forceps at 5 dpf, which was 1 day before the behavioral experiments. After surgery, the larvae were allowed to recover until the behavioral experiments. In the experiments of Fig. 2c, d, swim bladder-deflated fish were prepared as described below. Three to 4 hours before the

behavioral experiment, the fish was anesthetized using MS-222, and a glass pipette was inserted into the lateral–rostral–dorsal surface on the swim bladder where there was no PHM. The air in the swim bladder was then released. In the behavioral experiments, the fish was immersed in a 12.5% sucrose solution dissolved in fish-rearing water to compensate for the decrease in buoyancy.

### Head-restrained behavioral experiments

A larva at 6 dpf was embedded in 2% low-melting point agarose in a small acrylic chamber (12 [length] × 15 [width] × 4 [height] mm). Agarose located caudal to the swim bladder was removed, such that the fish was able to move the caudal body. The chamber was covered with a cover slip. The fish behaviors were filmed using the device shown in Supplementary Fig. 1i. The device was similar to that used to film head-free behaviors as described above, except that the setup for obtaining the frontal view was omitted. The head roll tilt stimulus was applied to the left- or right-down direction as described above. Images of the fish with the maximum bends are shown as examples of the fish's behavior.

### Dendra2 photo-conversion in TAN neurons

Five-dpf-fish of Tg(evx2:Gal4; UAS:Dendra2) were used for the photo-conversion experiment. A larva was anesthetized in 0.02% MS-222 and embedded in 2% low-melting point agarose on a glass bottom dish in the ventral-up position. The dish was placed under an inverted microscope (Leica microsystems, TCS SP8 MP). A 405-nm laser was applied to the prospective TAN neurons. The fish was kept for at least 3 hours to allow photoconverted Dendra2 protein to be transported to the axons, and confocal imaging was then performed.

### Electroporation

For the single-cell labeling by electroporation[60,61], a 6-dpf fish of Tg(evx2:GFP) was anesthetized in 0.02% MS-222. The fish was embedded in 3% low-melting point agarose dissolved in extracellular solution ((in mM) 134 NaCl, 2.9 KCl, 2.1 CaCl$_2$, 1.2 MgCl$_2$, 10 HEPES, and 10 glucose, 290 mOsm, adjusted to pH 7.8 with NaOH). Agarose covering the head was removed. The head skin was peeled off, and a thin layer of dorsal hindbrain was removed by suction through a glass pipette (diameter >10 μm) to allow to access to the tangential nucleus. Tetramethyl-rhodamine was electroporated into a single evx2:GFP-positive cell[61]. The fish was kept for at least 3 hours to allow the dye to be transported through the axon, and the electroporated cell was then imaged using an upright confocal microscope (Leica microsystems, TCS SP8 MP).

### Ca$^{2+}$ imaging setup

Ratiometric Ca$^{2+}$ imaging was performed using a tiltable objective microscope[20]. An objective lens and the fish were tilted using a motorized rotation stage (Thorlabs, DDR100/M). Blue light was shone onto the sample, and green (Cal-520 or GCaMP, Ca$^{2+}$-dependent signals) and red (tetramethyl-rhodamine dextran or tdTomato) signals were separated using imaging-splitting optics (Hamamatsu Photonics, W-View Gemini). Images of each channel were simultaneously recorded using a single digital camera at 10 fps (Hamamatsu Photonics, ORCA-Flash4.0 V3). For the imaging of nMLF neurons, a spinning-disk confocal scanner (Yokogawa, CSU-X1) was inserted into the illumination and detection light path (the same setup as that used previously[20]). For the imaging of TAN neurons and PHMs, a wide-field (non-confocal) version of a tiltable objective microscope (Supplementary Fig. 2d) was used. For this setup, the excitation light (Excelitas Technologies, X-Cite exacte) was delivered to the objective lens through an episcopic illuminator (Olympus, BX-URA2). The filters used in the episcopic illuminator were as follows: excitation, BP460-490 (Olympus); dichroic mirror, FF495-Di03 (Semrock); and emission, FF01-512/630 (Semrock). Fluorescence signals were

relayed to the imaging-splitting optics through a tube lens unit (Olympus, U-TR30-2) and a camera adaptor (Olympus, U-TV0.63XC). The filters used in the imaging-splitting optics were as follows: dichroic mirror, DM570 (Olympus); emission for the green channel, FF01-514/30 (Semrock); and emission for the red channel, BA575-625 (Olympus). The tilt stimuli were the same as those described in the behavioral experiments.

### Sample preparation for Ca$^{2+}$ imaging

Ca$^{2+}$ imaging was performed using 6 dpf fish with a nacre background[62]. A larval fish was mounted in 2% low-melting point agarose in an acrylic chamber (12 [length] × 15 [width] × 4 [height] mm). The chamber was then filled with fish-rearing water and covered with a fluorinated ethylene propylene (FEP) sheet with a refractive index of 1.34, which is close to that of water (1.33). The sample was viewed through the FEP sheet. The fish was mounted in the dorsal-up position for TAN and nMLF neuron imaging or in the ventral-up position for PHM imaging. For simultaneous imaging of Ca$^{2+}$ in PHMs and fish behaviors, agarose located at the caudal body was removed such that the fish was able to move the caudal part of its body.

### Ca$^{2+}$ imaging procedures

A fish was placed dorsal side up on a tiltable objective microscope. All images were recorded at 10 fps. For imaging of the TAN neurons in the Tg(evx2:tdTomato-jGCaMP7b) fish, Ca$^{2+}$ imaging was performed using a ×20/NA 0.5 water immersion lens (Olympus, UMPLANFL). TAN neurons were bilaterally illuminated using blue light (4.3 mW/mm$^2$ power at the sample) and imaged from the dorsal side. Occasionally, fast shifts of the image occurred during Ca$^{2+}$ imaging, presumably due to the occurrence of escape behaviors by the fish in agarose. The corresponding data were excluded from the analyses.

To image the nMLF, we retrogradely labeled them with 25% (w/v) Cal-520-dextran conjugate and 25% (w/v) dextran tetramethyl-rhodamine mixed in the extracellular solution. The mixture was injected into the second to fourth segments of the spinal cord in a 5 dpf larva using a tungsten pin. The larva was allowed to recover until 6 dpf. Ca$^{2+}$ imaging was performed with a spinning-disk confocal unit with a ×40/NA 0.8 objective lens (Olympus, LUMPLANFLN)[20]. A 488-nm laser (COHERENT, Sapphire 488-50 CDRH) was used as a light source with a laser power of 0.3–1.5 mW/mm$^2$ at the sample. nMLF neurons were imaged from the dorsal side. An imaging series was performed from the dorsal to the ventral plane and vice versa. For each fish, imaging of each plane was conducted only once owing to the desensitization of Cal-520.

In Ca$^{2+}$ imaging of PHMs, fish behaviors were monitored simultaneously (Fig. 8a, Supplementary Fig. 5c). Ca$^{2+}$ imaging and behavior imaging were performed from the ventral and dorsal sides, respectively. Tg(smyhc2:tdTomato-jGCaMP7b) fish were used to image the slow-type PHMs, and Tg(α-actin:tdTomato-jGCaMP7f) fish were used to image the fast-type PHMs. The muscle imaging was performed using a ×10/NA 0.3 lens (Olympus, UMPLANFLN). Illumination power was 4.2 and 5.7 mW/mm$^2$ for the slow- and fast-type PHM imaging, respectively. Fish behaviors were filmed at 100 fps using a camera (Teledyne FLIR, GS3-U3-23S6M) and a lens (Tamron, M118FM50) with infrared light illumination (Mightex, SLS-0208-E). To remove blue light in the behavioral images, a longpass filter (Olympus, BA610IF) was inserted into the optical path.

### Evaluation of a wide-field tiltable microscope

Evaluation of the tilt-derived (not neuronal activity-derived) $\Delta R/R_0$ in a wide-field version of the tiltable objective microscope was performed using fish that expressed green/red-Dendra2. Specifically, Dendra2-expressing TAN neurons in the Tg(evx2:Gal4; UAS:Dendra2) were briefly illuminated using violet light (400–420 nm) such that green-Dendra2 was partially photo-converted to red-Dendra2.

Observation of the TAN neurons during 20° tilts revealed that the amplitude of tilt-derived $\Delta R/R_0$ was confined within ±0.07 (Supplementary Fig. 2e, f).

To evaluate the artifacts derived from the tilt of a sample and muscle displacement in the PHM imaging, we used Tg(*pitx2*:Dendra2) fish that express Dendra2 at slow-type PHMs. Dendra2 was partially photo-converted, as described above. During tilt, maximum $\Delta R/R_0$ was mostly confined within 0.1, and occasionally up to 0.15 (Supplementary Fig. 5e–g).

### TAN neuron ablation
Laser ablation of TAN neurons was performed in Tg(*evx2*:GFP) larvae at 5 dpf. A larva was anesthetized in 0.02% MS-222 and embedded in 1.5% low-melting point agarose in the ventral-up position. Then, the fish was placed under a two-photon inverted microscope (Leica microsystems, TCS SP8 MP). All the GFP-labeled TAN neurons on one side (approximately 18 neurons) were subjected to laser ablation. The ablation was performed using a two-photon laser (wavelength 900 nm; InSight DeepSee 680–1300 nm) with a ×40/1.10 lens (Leica microsystems; No. 11506352 or 11506357). Scanning was immediately terminated when brief flashes of saturating intensity or a sudden decrease in transmitted light image intensity were observed. After laser ablation, the larva was allowed to recover and was fed until 6 dpf. Then, the larva was used for behavioral experiments. Successful ablations were verified after the behavioral experiments by checking for the absence of fluorescence. Fewer than four neurons survived. For a control ablation experiment, approximately 18 GFP-positive neurons located ventromedial to TAN neurons were subjected to laser ablation.

### Ablation of nMLF neurons
To ablate nMLF neurons, a combination of optical backfill and laser ablation was used (Fig. 7a). Homozygous Tg(*pitx2*:Dendra2) fish were used. First, a larva at 5 dpf was embedded lateral-side-down in 1.5% low-melting point agarose. Then the sample was placed under the inverted microscope (Leica microsystems, TCS SP8 MP). A 405-nm laser was applied from the third to ninth segments of the spinal cord using a ×20/0.75 lens (Leica microsystems, No. 11506344). Five to ten min of illumination was applied every 30 min for four times. After photo-conversion, the fish was removed from the agarose and kept for at least 4 hours under dark conditions. The larva was then re-embedded in the ventral-up position. Next, all the optically backfilled nMLF neurons, ~25 cells, were subjected to laser ablation. Observation of the fluorescence after the behavioral experiments revealed that fewer than six neurons survived. For a control ablation experiment, a similar number (~25) of Dendra2-positive neurons located in the hindbrain were subjected to laser ablation.

### Optogenetic activation experiments
Optogenetic activation of TAN neurons was performed in 6 dpf larvae of the Tg(*evx2*:CoChR-GFP) line. A larva was head-embedded in 2% low-melting point agarose in 35 mm glass bottom dish in the dorsal up position. Agarose located caudal to the swim bladder was removed, such that the fish was able to move the caudal body. The fish was placed under an upright microscope (Olympus, BX51WI) with a ×20/NA 0.5 water immersion objective lens (Olympus, UMPLANFLN). A 150-μm pinhole was inserted at the field-stop position, which allowed us to illuminate a target area that was ~50 μm in diameter. As a light source, a metal-halide lamp (Excelitas Technologies, X-Cite exacte) was used. The light was filtered using a bandpass filter FF02-482/18 (Semrock). Blue light illumination (2.0 mW/mm²) was delivered to the TAN neurons for 5 s. As a control illumination experiment, *evx2*-positive neurons located medial to the TAN neurons were illuminated. Fish behaviors were filmed from the bottom[32] at 50 fps using a ×2/ NA 0.14 objective lens (Olympus, XLFLUOR) and a camera (Teledyne FLIR, GS3-U3-23S6M) with FlyCature2 software (Teledyne FLIR).

### Immunohistochemistry
Immunostaining was conducted according to the Zebrafish Book (https://zfin.org/zf_info/zfbook/zfbk.html) and Doganli et al.[63]. The S58 monoclonal antibody supernatant (DSHB, RRID:AB_528377) was used as a primary antibody under 1:10 dilution conditions. Alexa fluor 488-conjugated goat anti-mouse IgG secondary antibody (Thermo Fisher Scientific, RRID:AB_2534088) was used at a 1:500 dilution.

### Labeling PHM-motoneurons by optical backfill
To label PHM-motoneurons (Supplementary Fig. 6d), a Tg(*vachta*:Gal4; UAS:Kaede) larva that was 5 dpf was used. The optical backfill was performed in the same way as the optical backfill of nMLF neurons that was described above. The illumination site of a 405-nm laser was the axons located in the middle and caudal segments of PHMs. After 3 to 4 hours, confocal imaging was performed.

### Data analysis
Images were processed using imageJ/Fiji, and numerical data were analyzed using Excel (Microsoft).

### Head-roll and body-bend angle
The head roll angle was measured as the angle between the horizontal line and the line connecting the top ends of both eyes. The frontal camera rotated together with the chamber, and therefore, the images were counter-rotated to cancel the rotation. To determine the top ends of both eyes, images were binarized, and the top position of each eye was defined as a mean position of the top 20 pixels. Trials in which the head roll angle before tilt was within ±10° were used in the analyses. To denoise, seven-frame moving averages were applied before the analyses. To calculate angular velocity of the counter-roll movements, angular velocity during a 1.5 s time window starting at the peak of the head roll was used.

The body bend angle in the head-free behavior experiment was measured as the angle between the following two lines: (1) line connecting the middle of both eyes and the caudal end of the swim bladder; and (2) line connecting the caudal end of the swim bladder and the tail end. Each point was separately detected by binarizing images. The caudal end of the swim bladder and the tail end were defined as the mean positions of ten and 20 pixels from edges, respectively. In the experiments where methylcellulose was used, the tail was not straight due to the high viscosity. Therefore, the middle point of the tail was used instead of the tail end to measure the body bend angle. The middle point of the tail was defined as half the length of the body from the swim bladder. To denoise, moving averages of seven frames were applied before the analyses. The body bend angle could not be accurately measured when the body was not viewed vertically. Thus, the measurements were terminated when the difference in tilt angle between the chamber and the head exceeded 10° (under these conditions, the dorsal view of the fish was too oblique to measure the bend angle).

In the behavioral experiments with the head embedded in agarose, the body bend angle was measured as described above. The tail end was determined as described above. The position of the middle of both eyes and the caudal end of the swim bladder did not change during the experiments. These two points were manually detected without binarization of the images. The angle before the roll tilts or photo-stimulation was converted to 0° and the body bend angle was shown as the angle difference compared to this default angle. To denoise, seven-frame moving averages were applied.

### Data analyses of Ca²⁺ imaging experiments
In the tiltable objective microscopes, the images rotated during the tilt. For registration purposes, green and red channel images were counter-rotated[20]. Then, the images were translated in the xy direction using Template Matching and Slice Alignment Plugin. Counter-rotation and

translation were performed with interpolation using the bilinear method. The $\Delta R/R_0$ of the region of interest in each frame was calculated, where $R_0$ was average ratio of the first ten frames. To denoise, seven-frame moving averages were applied, and the maximum $\Delta R/R_0 s$ were then calculated.

The positions of the nMLF neurons recorded from different fish were aligned in a horizontal plane using the midline (for medio-lateral axis) and the MeLm position (for rostro-caudal axis). When left and right MeLm neurons were labeled, the caudally located MeLm was used for the alignment. The recorded neurons were aligned along the dorso-ventral axis (z-axis) using the positions of MeLr and MeM as the landmarks. Four zones (z1 to z4) were defined as follows: z1, more dorsal than MeLr; z2, the dorsal half between MeLr and MeM; z3, the ventral half between MeLr and MeM; and z4, more ventral than MeM. The size of the soma area was measured in the plane where the soma was the largest.

## Statistics and reproducibility

For most of the experiments, multiple trials were performed in a fish. For the behavioral experiments in the methylcellulose solution, only one or two trials were performed in a fish. This was because the VBR performance worsened in repeated trials in the same fish, presumably due to fatigue of slow-type PHMs. For the $Ca^{2+}$ imaging in slow-type PHMs, one or two trials were analyzed in a fish. This was because trials during which bursts of rhythmic spontaneous activity occurred were excluded from the analysis. In the experiments where multiple trials were performed in a fish, the mean value was calculated, and it was used as a representative result for the fish. In the experiments of Figs. 4a, 9a and Supplementary Figs. 2c, 4, 5a, 5b, 5d, 6d, more than two individual experiments were performed, and results that were similar to the representative images were observed across animals.

Statistical analysis was performed using R. The statistical significance was assessed using paired samples $t$-test, two-sample $t$ test, Wilcoxon Exact rank-sum test, or the Steel Dwass test. All tests were two-tailed tests. Before performing paired samples $t$ test, the normality of the distribution for the difference values between paired samples was confirmed using the Shapiro–Wilk test. For two-sample $t$-test, the normality of the distribution and the variance were checked using the Shapiro–Wilk test and $F$ test, respectively. Statistical results were indicated as follows: *, $p$ value <0.05; **, $p$ value <0.01; ***, $p$ value <0.001; or N.S. (not significant), $p$ value ≥0.05.

## Reporting summary

Further information on research design is available in the Nature Portfolio Reporting Summary linked to this article.

## Data availability

Source data are provided with this paper.

## Code availability

Code used for measurements of head roll/body bend angles and the image registration are available in Zenodo with identifier https://doi.org/10.5281/zenodo.7583110.

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

## Acknowledgements

We thank Yoichi Oda for critical reading of the manuscript; Minoru Koyama for sharing CoChR-GFP-Kv2.1 plasmid; Higashijima lab members for their help with generating transgenic fish, fish care, and discussion; and the Optics and Imaging Facility of the National Institute for Basic Biology for the use of their confocal microscopes. This work was supported in part by grants from the Ministry of Education, Culture, Sports, Science and Technology of Japan and from National Bioresource Project in Japan (KAKENHI Grant Numbers JP18KK0215, JP19H03333, and JP22H02666 to S.H., and JP20K06866 to M.T.,) and MEXT National BioResource Project (NBRP) to S.H.

## Author contributions

T.S., M.T., and S.H. designed the research; T.S. performed most of the experiments under the supervision of M.T.; S.H. generated transgenic fish; and T.S. analyzed the data. T.S., M.T., and S.H. wrote the manuscript.

## Competing interests

The authors declare no competing interests.
