## [Peer Review File · Nature Communications]

Biomechanics and neural circuits for vestibular-induced fine postural control in larval zebrafishREVIEWER COMMENTS

Reviewer #1 (Remarks to the Author):

Sugioka and colleagues present a clever, innovative, and exciting set of experiments to define the neural mechanisms governing postural control in the larval zebrafish. Their results clearly lay out a pathway from tangential nucleus to nucleus MLF to a specialized ventral (hypaxial) muscle in the rostral spinal cord. The authors demonstrate that each of these elements in turn is important to the postural correction. These results also bring together several individual but weaker pieces of evidence previously shown in the literature to make a much more convincing case for the importance of this pathway in postural control. The writing is clear and succinct. The work provides strong evidence for the significance of this pathway to postural control. It is exciting, uses creative and cutting-edge techniques, and will open up a series of further interesting questions in the coordination of turning and posture. These types of analyses are very difficult to carry out in other models right now, and therefore this work should be influential.

Major comments

--I was not able to find statistical analysis of the behavior presented in Fig. 1. While the presented traces are compelling, the claims must be supported statistically (eg the loss of postural control in Fig. 1k). If I have missed the analysis, an additional reference in the text or figure legend to it would be helpful.

--The data show that ablation of the nMLF neurons reduces, but does not abolish, the right response. One possible alternate pathway is via direct connections from the tangential nucleus. The tangential nucleus neurons often bifurcate, sending a caudal-projecting portion of their axon. Can the authors speculate on why the vestibular system might use the somewhat indirect route they propose here (via nMLF) rather than simply having synaptic drive directly from the tangential descending neurons onto the posterior hypaxial motor neurons? Or is that descending pathway likely responsible for the residual postural behavior seen in Fig 6e? Unless the authors have reason to rule out this descending component of control, it should perhaps be included as a dashed line or speculative connection in their schematic in Fig 2.

Minor comments

--The authors show that the posterior hypaxial muscle is activated by ipsilateral-up tilts. It is interesting to see, however, that there also appears to be some activation during ipsilateral-down tilts (Fig 7c, d). The authors should quantify whether this activation is significantly different from zero. I do not think it

poses a problem for their model, as there is clear a large asymmetry between the two sides, but it would be interesting to know if it is “real”. If it is a significant increase, the authors should comment on the possible explanations in the Discussion. For example, could there be a slight ventroflexion during posture correction? Or could it be that both sides of the body are activated, just asymmetrically? (as per Bagnall & McLean 2014).

Reviewer #2 (Remarks to the Author):

The authors present a systematic analysis of the pathway through which larval zebrafish perform fine axial control, using vestibular information to correct for displacement in the roll axis. The study is impressive for its capable use of several diverse methods, and experiments are well conceived, well controlled, and interpretable. This is high-quality work.

Its shortcoming is that there is little in the way of conceptual advances, with most results corroborating previous studies. The proposed pathway is not novel, and the manuscript’s main contribution is to perform functional confirmations of the individual elements’ roles in delivering behavior. As such, I view the advance as incremental. Below I illustrate this view by listing previous findings, alongside those that are novel to this manuscript.

Beyond this major concern, I find very little to criticize. The literature review, writing, figures, experimental design, and interpretations are all sound. The manuscript contains a particularly enjoyable discussion section, touching on buoyancy and stability more broadly, and nicely summarizing the evolutionary conservation of the fine axial control circuits.

Observations that have been made previously:

Vestibular signals produce a body bend near the swim bladder (named the VBR in this manuscript). The bend is graded to stimulus strength and lasts as long as the off-center stimulus persists.

Strong stimuli cause active (swimming) correction, where weak stimuli cause only the fine axial response (VBR).

TAN nucleus activation.

TAN projections to the nMLF.

Unilateral activation of the nMLF in response to a fictive roll stimulus.

VBR in response to optogenetic nMLF activation.

Involvement of PHMs in the nMLF-mediated body bend.

Novel observations in this manuscript:

A buoyancy-based mechanism for VBR is proposed and tested.

TAN ablation reduces VBR. TAN activation drives VBR.

The nMLF neurons activated are among the small neurons of the nMLF.

Unilateral ablation of nMLF neurons causes a unilateral impairment of VBR.

Slow-type PHM activity correlates with VBR, and slow PHM ablation reduces VBR.

Reviewer #3 (Remarks to the Author):

The manuscript by Sugioka et al explores both the biomechanical and circuit basis for fine postural control in larval zebrafish. Previous work studying postural control in fish has focused on dynamic adjustments that occur during swimming. However, it has been an open question whether fish would have a fine postural control system akin to land vertebrates, where anti-gravity demands are more obvious. The authors now demonstrate this is indeed the case, using an impressive combination of modeling, behavioral analysis, calcium imaging, neuronal ablations and optogenetics, which ultimately demonstrate the necessity and sufficiency of many of the circuit components. Specifically, the study identifies a highly conserved pathway from sensation to action, where roll tilt is detected by tangential neurons in the vestibular nucleus, which relay information to excitatory neurons in a midbrain reticulospinal nucleus, which contribute excitation of a specialized class of muscles responsible for subtle bends in the rostral trunk. Virtually identical pathways have been described in mammals, and so the findings are likely broadly applicable. The manuscript is well written, the figures are beautiful, and the discoveries are likely to make a big impact in the field, as the Higashijima lab continues to identify core principles of circuit operation with respect to motor control in zebrafish. The work helps to reconcile previous observations and also opens up a number of new avenues of investigation with respect to both dynamic and static control of posture. I have only a few minor comments.

Minor comments

- 1) It is not clear from the first experiment how the fish are restrained. Are they simply in a fluid filled chamber? Or is there any agarose in there? Apologies if I missed something here.
- 2) It would be nice to move as much of the supplemental information to the main figures as possible. I found myself having to dive into the supplemental text too often to track down pretty important information. For instance, many important controls are dumped in supplemental, in addition to the optogenetic experiments demonstrating sufficiency (Fig. 3). It looks like there is space in the main figures to add panels.

We are grateful for the reviewers' thoughtful comments and suggestions. As a result, we have made substantial revisions to our manuscript, adding new experimental data, figure panels, statistical analyses, and discussion. We have also moved some of the supplemental figures to the main figures. We believe that the revisions have increased the readability and impact of the manuscript. We have addressed all of the reviewer comments, as described below. The reviewer comments are colored blue and italicized.

REVIEWER COMMENTS

Reviewer #1:

Sugioka and colleagues present a clever, innovative, and exciting set of experiments to define the neural mechanisms governing postural control in the larval zebrafish. Their results clearly lay out a pathway from tangential nucleus to nucleus MLF to a specialized ventral (hypaxial) muscle in the rostral spinal cord. The authors demonstrate that each of these elements in turn is important to the postural correction. These results also bring together several individual but weaker pieces of evidence previously shown in the literature to make a much more convincing case for the importance of this pathway in postural control. The writing is clear and succinct. The work provides strong evidence for the significance of this pathway to postural control. It is exciting, uses creative and cutting-edge techniques, and will open up a series of further interesting questions in the coordination of turning and posture. These types of analyses are very difficult to carry out in other models right now, and therefore this work should be influential.

We are grateful for these kind comments.

Major comments

I was not able to find statistical analysis of the behavior presented in Fig. 1. While the presented traces are compelling, the claims must be supported statistically (eg the loss of postural control in Fig. 1k). If I have missed the analysis, an additional reference in the text or figure legend to it would be helpful.

Based on this useful suggestion, we performed statistical analyses to compare the recovery performances (head roll angles at 4 s) of the intact, pectoral fin-removed, and swim bladder-deflated fish. The result is presented in a newly generated Supplementary figure 1h and described in the text (line 120 and 189).

The data show that ablation of the nMLF neurons reduces, but does not abolish, the right response. One possible alternate pathway is via direct connections from the tangential nucleus. The tangential nucleus neurons often bifurcate, sending a caudal-projecting portion of their axon. Can the authors speculate on why the vestibular system might use the somewhat indirect route they propose here (via nMLF) rather than simply having synaptic drive directly from the tangential descending neurons onto the posterior hypaxial motor neurons? Or is that descending pathway likely responsible for the residual postural behavior seen in Fig 6e? Unless the authors have reason to rule out this descending component of control, it should perhaps be included as a dashed line or speculative connection in their schematic in Fig 2.

There are two possibilities for why the ablation of nMLF neurons did not completely abolish the VBR response. The first possibility is that not all nMLF neurons were ablated. Because of this, the residual nMLF neurons contributed to producing the weakened VBR. The second possibility is that the descending axons of the TAN neurons contribute to the VBR, as the reviewer suggested. Our anatomical study revealed that the descending axons of the TAN neurons project to the rostral spinal cord, but they did not appear to reach the somata of PHM motoneurons (Fig. 3a, Supplementary Fig. 6d). Thus, even if the descending axons contribute to the VBR, they would exert the action via a polysynaptic pathway. We have added these sentences to the Discussion (line 478-485). As for your suggestion to draw a dashed line in the schematic in Figure 2, we have avoided doing this because Figure 2 is presented to show the pathway that we will examine in the rest of the manuscript (the role of the descending axons of the TAN neurons is not examined in the present study).

As for the reason why the vestibular system uses a somewhat indirect route (via nMLF) to produce the VBR, there is no clear answer. One potential advantage, we think, would be that the involvement of nMLF neurons in the pathway would give flexibility in the circuits. For example, in the situation when the head roll is larger (e.g., 45 deg), fish may prefer dynamic control (via swimming) to fine control (via VBR) for postural correction. The involvement of nMLF neurons in the VBR pathway would make such switching easier, as nMLF neurons are also involved in swimming behavior. If the VBR is solely controlled via the descending axons of the TAN neurons, such flexibility would not exist, and the VBR would occur regardless of the state of the fish. This is speculation, however, and as such, we have avoided discussing this.

Minor comments

The authors show that the posterior hypaxial muscle is activated by ipsilateral-up tilts. It is interesting to see, however, that there also appears to be some activation during ipsilateral-down tilts (Fig 7c, d). The authors should quantify whether this activation is significantly different from zero. I do not think it poses a problem for their model, as there is clear a large asymmetry between

the two sides, but it would be interesting to know if it is “real”. If it is a significant increase, the authors should comment on the possible explanations in the Discussion. For example, could there be a slight ventroflexion during posture correction? Or could it be that both sides of the body are activated, just asymmetrically? (as per Bagnall & McLean 2014).

In Ca²⁺ imaging of the PHMs, artificial fluorescent changes can be derived not only from the tilt of a sample but also from muscle displacement during the VBR. Therefore, we performed a new experiment to estimate the level of artifacts during PHM imaging using Tg(*pitx2:Dendra2*) fish that expressed green/red Dendra2 in slow-type PHMs (Supplementary Fig. 5d, e). During tilt, the maximum $\Delta R/R_0$ was mostly confined within 0.1 (Supplementary Fig. 5f, g). Taking this into account, PHMs are judged to be occasionally (but not always) active during ipsilateral-down tilts (Fig. 7c, d). We have added a description of this to line 369.

As the reviewer mentioned, bilateral activations of PHMs may cause a slight ventroflexion. Theoretically, this ventroflexion could contribute to recovering the upright posture, as this would result in a ventral shift of the COM in the cross section near the COM (downward and slightly rightward shift of the COM in the right bottom panel of Fig. 1h). The consequence would be an increase in the distance between the COM and the COV, which increases the moment of force. However, two lines of experimental data suggest that the ventroflexion, if any, is not salient. First, as noted above, PHMs were not always activated during ipsilateral-down tilts. In the rostral PHMs, for example, only two out of six fish exhibited the activities during ipsilateral-down tilts (Fig. 7d). Second, no obvious downward movement of the head was observed in the fish behavior in the experiment using methylcellulose solution (frontal view in Supplementary movie 3). We have checked more than 10 trials. Except for one trial, no obvious downward movement of the head was observed. Although a subtle ventroflexion cannot be reliably quantified from the frontal view, our observations suggest that the ventroflexion, if any, is very small.

In summary, available experimental data do not strongly support ventroflexion. Thus, both sides of the body are likely activated just asymmetrically. The physiological significance of the PHM activity on the ear-down side is currently unknown. As such, we have avoided discussing the PHM activity on the ear-down side.

Reviewer #2:

The authors present a systematic analysis of the pathway through which larval zebrafish perform fine axial control, using vestibular information to correct for displacement in the roll axis. The study is impressive for its capable use of several diverse methods, and experiments are well conceived, well controlled, and interpretable. This is high-quality work.

Its shortcoming is that there is little in the way of conceptual advances, with most results

corroborating previous studies. The proposed pathway is not novel, and the manuscript's main contribution is to perform functional confirmations of the individual elements' roles in delivering behavior. As such, I view the advance as incremental. Below I illustrate this view by listing previous findings, alongside those that are novel to this manuscript.

Beyond this major concern, I find very little to criticize. The literature review, writing, figures, experimental design, and interpretations are all sound. The manuscript contains a particularly enjoyable discussion section, touching on buoyancy and stability more broadly, and nicely summarizing the evolutionary conservation of the fine axial control circuits.

Observations that have been made previously:

Vestibular signals produce a body bend near the swim bladder (named the VBR in this manuscript). The bend is graded to stimulus strength and lasts as long as the off-center stimulus persists.

Strong stimuli cause active (swimming) correction, where weak stimuli cause only the fine axial response (VBR).

TAN nucleus activation.

TAN projections to the nMLF.

Unilateral activation of the nMLF in response to a fictive roll stimulus.

VBR in response to optogenetic nMLF activation.

Involvement of PHMs in the nMLF-mediated body bend.

Novel observations in this manuscript:

A buoyancy-based mechanism for VBR is proposed and tested.

TAN ablation reduces VBR. TAN activation drives VBR.

The nMLF neurons activated are among the small neurons of the nMLF.

Unilateral ablation of nMLF neurons causes a unilateral impairment of VBR.

Slow-type PHM activity correlates with VBR, and slow PHM ablation reduces VRB.

We appreciate the reviewer's evaluation of our manuscript as a high-quality work. We agree with all of the listed strengths and weaknesses of our manuscript. As the reviewer noted, one of the novelties of the present work is a buoyancy-based mechanism for postural control, which we think is a conceptual advance in the biomechanics of posture control in the aquatic environment. We were pleased to read the following statement: "*The manuscript contains a particularly enjoyable discussion section, touching on buoyancy and stability more broadly, and nicely summarizing the evolutionary conservation of the fine axial control circuits.*" We believe that readers in the broad community will enjoy reading our manuscript upon publication in *Nature Communications*.

Reviewer #3:

The manuscript by Sugioka et al explores both the biomechanical and circuit basis for fine postural control in larval zebrafish. Previous work studying postural control in fish has focused on dynamic adjustments that occur during swimming. However, it has been an open question whether fish would have a fine postural control system akin to land vertebrates, where anti-gravity demands are more obvious. The authors now demonstrate this is indeed the case, using an impressive combination of modeling, behavioral analysis, calcium imaging, neuronal ablations and optogenetics, which ultimately demonstrate the necessity and sufficiency of many of the circuit components. Specifically, the study identifies a highly conserved pathway from sensation to action, where roll tilt is detected by tangential neurons in the vestibular nucleus, which relay information to excitatory neurons in a midbrain reticulospinal nucleus, which contribute excitation of a specialized class of muscles responsible for subtle bends in the rostral trunk. Virtually identical pathways have been described in mammals, and so the findings are likely broadly applicable. The manuscript is well written, the figures are beautiful, and the discoveries are likely to make a big impact in the field, as the Higashijima lab continues to identify core principles of circuit operation with respect to motor control in zebrafish. The work helps to reconcile previous observations and also opens up a number of new avenues of investigation with respect to both dynamic and static control of posture. I have only a few minor comments.

We are grateful for the kind comments.

Minor comments

1) It is not clear from the first experiment how the fish are restrained. Are they simply in a fluid filled chamber? Or is there any agarose in there? Apologies if I missed something here.

The fish are simply in a fluid-filled chamber. In the revised manuscript, we state this (line 93).

2) It would be nice to move as much of the supplemental information to the main figures as possible. I found myself having to dive into the supplemental text too often to track down pretty important information. For instance, many important controls are dumped in supplemental, in addition to the optogenetic experiments demonstrating sufficiency (Fig. 3). It looks like there is space in the main figures to add panels.

Based on the suggestions, we have moved the following figure panels to the main figures: 1) control ablation experiments for TAN neurons (Figs. 4e-h); 2) optogenetic activation experiments for TAN neurons (Figs. 4i-p); and 3) control ablation experiments for nMLF neurons (Figs. 6f-i). Upon moving these figure panels to the main figures, we noticed that the number of fish in some of the control experiments was small. Consequently, we performed additional experiments to increase the number of fish and the results are presented in the new figure panels (Figs. 4h, 4o, and 6i). As for the optogenetic activation of TAN neurons, we changed the presentation style. In the previous figure, the body bend angles before and during the photo-stimulation were compared. In the revised figure (Fig. 4o), the body bend angles of the experimental and control animals during the photo-stimulation are compared. This change was made because the large difference in the body angles between the experimental and control animals is the point that we would like to maintain.

REVIEWERS' COMMENTS

Reviewer #1 (Remarks to the Author):

The authors have adequately addressed my review comments and I have no further concerns.

Reviewer #2 (Remarks to the Author):

The revised manuscript appears to address the major concerns of Reviewer 1, although I defer to him/her on whether these changes are satisfactory. I continue not to be convinced that the advance in this manuscript is sufficient for Nature Communications, but I recognise and respect the enthusiasm of the two other experts, and endorse publication on this basis.

Reviewer #3 (Remarks to the Author):

Thank you, the authors have fully addressed my comments.

REVIEWER COMMENTS

Reviewer #1 (Remarks to the Author):

The authors have adequately addressed my review comments and I have no further concerns.

Reviewer #2 (Remarks to the Author):

The revised manuscript appears to address the major concerns of Reviewer 1, although I defer to him/her on whether these changes are satisfactory. I continue not to be convinced that the advance in this manuscript is sufficient for Nature Communications, but I recognise and respect the enthusiasm of the two other experts, and endorse publication on this basis.

Reviewer #3 (Remarks to the Author):

Thank you, the authors have fully addressed my comments.

Thank you for reviewing our revised manuscript. We are grateful for your helpful comments and suggestions.